# BLADE: A DERIVATIVE-FREE BAYESIAN INVERSION METHOD USING DIFFUSION PRIORS

## ABSTRACT

Derivative-free Bayesian inversion is an important task in many science and engineering applications, particularly when computing the forward model derivative is computationally and practically challenging. In this paper, we introduce `Blade`, which can produce accurate and well-calibrated posteriors for Bayesian inversion using an ensemble of interacting particles. `Blade` leverages powerful data-driven priors based on diffusion models, and can handle nonlinear forward models that permit only black-box access (i.e., derivative-free). Theoretically, we establish a non-asymptotic convergence analysis to characterize the effects of forward model and prior estimation errors. Empirically, `Blade` achieves superior performance compared to existing derivative-free Bayesian inversion methods on various inverse problems, including challenging highly nonlinear fluid dynamics.

## 1 INTRODUCTION

Inverse problems, which seek to infer underlying system states or parameters from indirect and noisy observations, are fundamental to numerous applications in science and engineering. For instance, numerical weather prediction requires inferring the atmospheric state by assimilating observational data from weather stations and satellites (Bannister, 2017). Solving these problems requires overcoming three major challenges: first, they are often high-dimensional and ill-posed, meaning that the solution may be non-unique or unstable under perturbations (Hadamard, 2014); second, the design of priors or regularizers is non-trivial and has a significant impact on the solution; third, the associated forward models may involve complicated numerical algorithms that make derivative calculation impractical. Indeed, the weather example admits multiple possible solutions, demands carefully designed priors, and involves intricate numerical procedures with various **non-differentiable** steps like remapping, branching, and discrete search (Park & Xu, 2013; White, 2000). As such, derivative-free Bayesian inversion methods that use flexible priors to perform reliable uncertainty quantification for high dimensional problems are quite desirable (Park & Xu, 2013).

We consider the inverse problems in the canonical form:

$$\boldsymbol{y} = \mathcal{G}(\boldsymbol{x}^*) + \boldsymbol{\epsilon}, \tag{1}$$

where $\boldsymbol{y} \in \mathbb{R}^m$ is the observation or measurement, $\boldsymbol{x}^* \in \mathbb{R}^n$ is the unknown state, $\mathcal{G}$ is the forward model accessible only via forward evaluations (i.e., black-box access), and $\boldsymbol{\epsilon} \in \mathbb{R}^m$ is the measurement noise, often modeled as additive Gaussian $\mathcal{N}(0, \sigma_y^2\mathbf{I})$. The Bayesian framework characterizes the solution as a posterior distribution $p(\boldsymbol{x}^*|\boldsymbol{y}) \propto p(\boldsymbol{x}^*)p(y|\boldsymbol{x}^*)$ that enables uncertainty quantification for principled decision-making (Sanz-Alonso et al., 2023). As the gradient of $\mathcal{G}$ is difficult or impractical to compute, one typically resorts to derivative-free Bayesian inversion methods.

Traditional methods for derivative-free Bayesian inversion include Markov chain Monte Carlo (MCMC) methods (Geyer, 1992; Gelman et al., 1997; Cotter et al., 2013) and Sequential Monte Carlo (SMC) (Del Moral et al., 2006). These methods offer convergence guarantees but face significant scalability challenges for high dimensional problems. Approximate Bayesian methods (Garbuno-Inigo et al., 2020a; Carrillo et al., 2022; Huang et al., 2022a) offer better efficiency, but often struggle to capture complex posteriors. Additionally, these methods require access to the prior density (up to a normalizing constant), which can be difficult to directly model in high dimensions.

Many recent derivative-free algorithms (Zheng et al., 2025a; Tang et al., 2024; Huang et al., 2024) leverage diffusion models (DMs) as plug-and-play priors for solving high-dimensional inverse

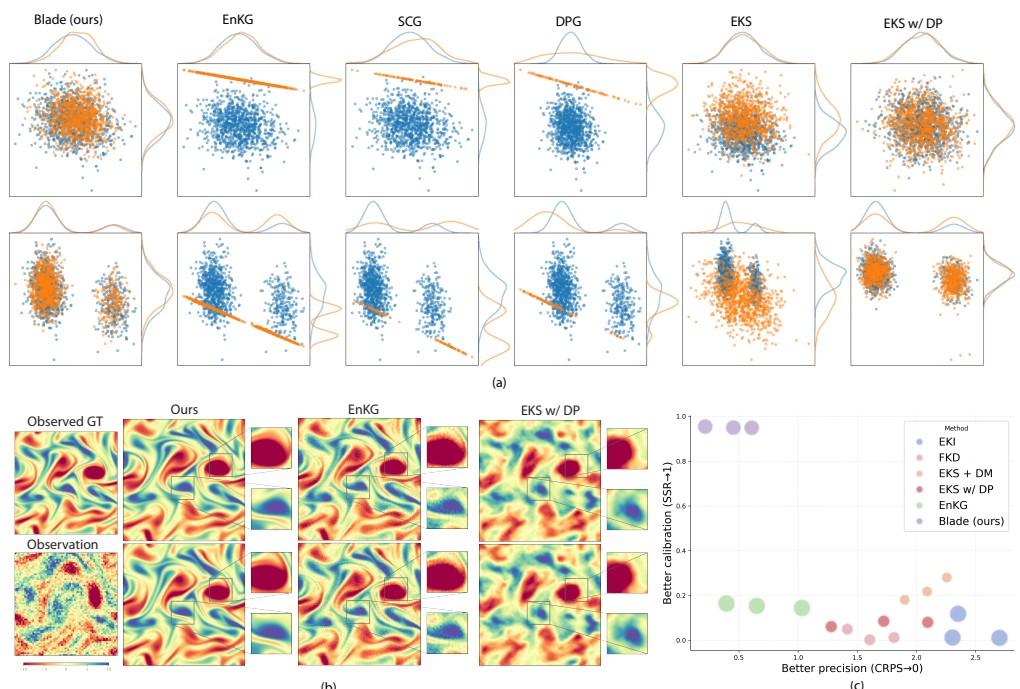

Figure 1: (a): Results on linear Gaussian and Gaussian mixture problems. Blue samples are from the ground-truth posterior. (b): Posterior draws from different methods on Navier-Stokes problem. "Observed GT" marks a single observed ground truth. `Blade` produces smooth, structured samples with realistic variability, while the competing methods yield noisier samples that stuck in a single blurred mode. See detailed comparison in Fig. 9. (c): CRPS (continuous ranked probability score) versus SSR (spread-skill ratio) under varying measurement noise levels, with area indicating the relative runtime cost. Only `Blade` produces well-calibrated samples among derivative-free methods.

problems with complex prior distributions. DMs can flexibly capture complex prior distributions from data, but require optimization or sampling for posterior inference, mainly due to modeling the score function rather than the density. Optimization-based approaches disregard posterior spread and thus often fail to capture spread or uncertainty even in simple Gaussian settings with linear forward models (see Fig. 1). Sampling-based derivative-free approaches can be asymptotically correct (Trippe et al., 2022; Wu et al., 2023; Cardoso et al., 2024; Dou & Song, 2024), but have not yet demonstrated the capability to produce probabilistically calibrated samples in the high-dimensional nonlinear setting.

Our contributions are summarized as follows:

- We propose `Blade`, a derivative-free, ensemble-based Bayesian inversion algorithm that can produce well-calibrated posterior samples for inverse problems with diffusion prior.
- We establish convergence analysis and provide explicit error bounds to quantify the impact of score approximation and statistical linearization error on sampling quality.
- We evaluate `Blade` through various probabilistic verifications. In controlled settings we perform direct distributional checks against the ground truth posterior. In a challenging nonlinear fluid dynamics problem, we assess posterior quality using standard probabilistic metrics. Across all these tests, `Blade` demonstrates superior performance compared to competing approaches.

## 2 BACKGROUND

**Diffusion Models.** We consider diffusion models in the unified EDM framework (Karras et al., 2022). Diffusion models define a forward stochastic process to evolve the original data distribution $p_0(\boldsymbol{x})$ to an approximately Gaussian distribution $p_T(\boldsymbol{x}) = \mathcal{N}(0, s^2(T)\sigma(T)^2\mathbf{I})$, where $\sigma(t)$ is a pre-defined noise schedule function and $s(t)$ is the pre-defined scaling function. Without loss of generality, we set $s(t) = 1$ because every other schedule is equivalent to it up to a simple reparameterization as

shown in Karras et al. (2022). We consider the following form of denoising diffusion process:

$$d\boldsymbol{x}_t = -\left(2\dot{\sigma}(t)\sigma(t) + \beta(t)\right)\nabla_{\boldsymbol{x}_t}\log p\left(\boldsymbol{x}_t; \sigma(t)\right)dt + \left(\sqrt{2\dot{\sigma}(t)\sigma(t) + 2\beta(t)}\right)d\bar{\boldsymbol{w}}_t, \quad (2)$$

where $\beta(t)$ can be any non-negative function as shown in Zhang & Chen (2022). Generating new samples from $p_0(\boldsymbol{x})$ amounts to integrating Eq. (2) from a random sample from $p_T(\boldsymbol{x}_T)$. This requires computation of the time-dependent score function $\nabla\log p(\boldsymbol{x}_t; \sigma(t))$, which can be approximated with a neural network: $s_\theta(\boldsymbol{x}_t, t) \approx \nabla\log p(\boldsymbol{x}_t; \sigma(t))$. In our work, we assume that we have access to such a pre-trained score function, which we will simply refer to as the diffusion model.

**Split Gibbs Sampling.** The Split Gibbs Sampler (SGS) (Vono et al., 2019) is a Markov chain Monte Carlo (MCMC) method that aims to sample the posterior $p(\boldsymbol{x} \mid \boldsymbol{y}) \propto p(\boldsymbol{y} \mid \boldsymbol{x})p(\boldsymbol{x}) = \exp(-f(\boldsymbol{x}; \boldsymbol{y}) - g(\boldsymbol{x}))$, where $f(\boldsymbol{x}; \boldsymbol{y}) = -\log p(\boldsymbol{y} \mid \boldsymbol{x})$ and $g(\boldsymbol{x}) = -\log p(\boldsymbol{x})$. Instead of direct posterior sampling, SGS samples the auxiliary distribution:

$$\pi^{XZ}(\boldsymbol{x}, \boldsymbol{z}) \propto \exp\left(-f(\boldsymbol{z}; \boldsymbol{y}) - g(\boldsymbol{x}) - \frac{1}{2\rho^2}\|\boldsymbol{x} - \boldsymbol{z}\|_2^2\right), \quad (3)$$

where $\boldsymbol{z} \in \mathbb{R}^n$ is an auxiliary variable and $\rho$ is a parameter that controls the distance between $\boldsymbol{x}$ and $\boldsymbol{z}$. As shown by Vono et al. (2019), sampling the posterior becomes equivalent to sampling Eq. (3) as $\rho$ approaches 0. Suppose $\boldsymbol{x}^{(0)}$ is the initial state and $k$ is iteration index. Sampling Eq. (3) is achieved using a Gibbs sampling procedure that alternates between the following two steps:

$$\boldsymbol{z}^{(k)} \sim \pi^{Z|X=\boldsymbol{x}^{(k)}}(\boldsymbol{z}) \propto \exp\left(-f(\boldsymbol{z}; \boldsymbol{y}) - \frac{1}{2\rho^2}\left\|\boldsymbol{x}^{(k)} - \boldsymbol{z}\right\|_2^2\right) \quad \text{(likelihood step)} \quad (4)$$

$$\boldsymbol{x}^{(k+1)} \sim \pi^{X|Z=\boldsymbol{z}^{(k)}}(\boldsymbol{x}) \propto \exp\left(-g(\boldsymbol{x}) - \frac{1}{2\rho^2}\left\|\boldsymbol{x} - \boldsymbol{z}^{(k)}\right\|_2^2\right) \quad \text{(prior step)} \quad (5)$$

**Diffusion-based Split Gibbs Sampling.** Recent works have explored adapting generative model priors into the Gibbs sampling (Janati et al., 2025; Achituve et al., 2025) and split Gibbs framework (Bouman & Buzzard, 2023; Coeurdoux et al., 2023; Xu & Chi, 2024; Wu et al., 2024; Chu et al., 2025). These methods focus on developing different ways to realize the prior step. This is orthogonal to our `Blade` method, which primarily focuses on the design of the likelihood step. For the prior step, we follow the practice of Wu et al. (2024) which deduces the prior step to a denoising diffusion process within the EDM framework (Karras et al., 2022), enabling the use of any pre-trained diffusion model as the generative prior. For the likelihood step, the prior work either uses gradient-based Langevin Monte Carlo or derives the closed-form expression of $\pi^{Z|X}$ for linear problems, which requires the access to the gradient or adjoint operator of the forward model. In contrast, our `Blade` method implements the likelihood step as an interacting particle system which does not rely on derivative or adjoint operator information.

**Ensemble Kalman Methods.** Ensemble Kalman methodology was first introduced by Evensen (1994) as a way of performing statistical linearization (Booton, 1954): estimating a surrogate linear model from samples (e.g., those generated using a black-box forward model). This approach is appealing since it sidesteps the need to compute gradients from the generating process, and can then be integrated into various Bayesian inference schemes. In the inverse problem setting, this idea was used to develop optimization-based frameworks such the Ensemble Kalman Inversion (EKI) framework (Iglesias et al., 2013; Kovachki & Stuart, 2018; Iglesias, 2016; Chada et al., 2020; Huang et al., 2022b) as well as derivative-free diffusion guidance methods (Kim et al., 2024; Zheng et al., 2025a). Our `Blade` method also builds upon the fundamental idea of statistical linearization of a black-box forward model, and carefully incorporates it into a split Gibbs framework with a diffusion prior to generate well-calibrated posterior samples for high-dimensional inverse problems.

## 3 BLADE METHOD

The name `Blade` is derived from the key components of the algorithm: Bayesian inversion, Linearization, Alternating updates, Derivative-free, and Ensembling. As illustrated in Fig. 2 and summarized in Alg. 1, `Blade` is built on the split Gibbs framework and evolves a set of interacting particles that alternate between a derivative-free likelihood sampling step and a denoising diffusion prior step. In

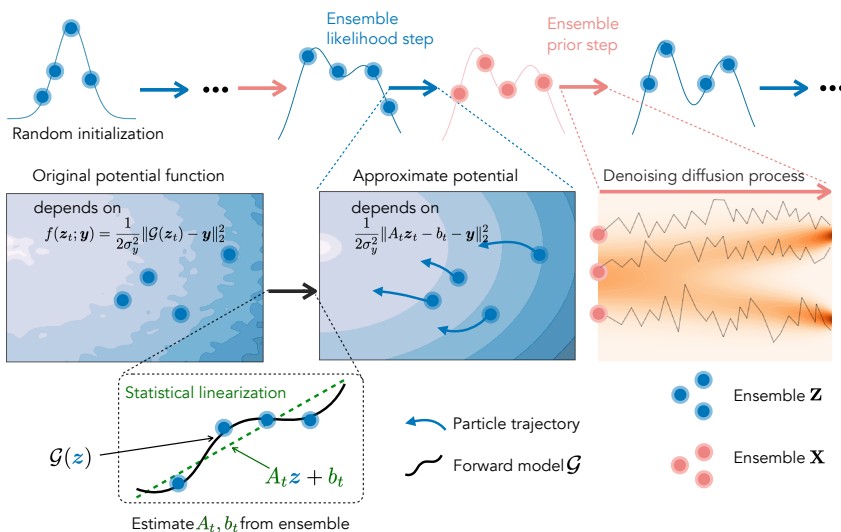

Figure 2: Illustrative depiction of `Blade` (see Sec. 3).

---

**Algorithm 1** `Blade` method for derivative-free Bayesian inversion using diffusion priors

---

**Require:** initial ensemble $\mathbf{X}^{(0)} = \{\mathbf{x}^{(j)} \in \mathbb{R}^n\}_{j=1}^J$, number of iterations $K$, $\{\rho_k\}_{k=0}^K$, observed data $\mathbf{y} \in \mathbb{R}^m$, pre-trained diffusion model $s_\theta$

1: **for** $k \in \{0, \dots, K-1\}$ **do**
2:     $\mathbf{Z}^{(k)} \leftarrow$ Ensemble-likelihood-step$(\mathbf{X}^{(k)}, \rho_k)$           $\triangleright$ Algorithm 2
3:     $\mathbf{X}^{(k+1)} \leftarrow$ Ensemble-prior-step$(\mathbf{Z}^{(k+1)}, s_\theta, \rho_k)$      $\triangleright$ Algorithm 3
4: **end for**
5: **return** $\mathbf{X}^{(K)}$

---

the likelihood step, the original (and potentially complex) potential is approximated by a smooth quadratic form through statistical linearization, which enables derivative-free sampling (Sec. 3.1). In the prior step, the ensemble member operates independently following denoising diffusion process (Sec. 3.2). Sec. 3.3 presents the full algorithm along with practical considerations. Finally, Sec. 3.4 provides a non-asymptotic theoretical analysis, quantifying the approximation errors introduced by statistical linearization and the learned prior score.

## 3.1 DERIVATIVE-FREE LIKELIHOOD STEP VIA STATISTICAL LINEARIZATION

Let $\mathbf{X}^{(k)} = \{\boldsymbol{x}^{(j)}\}_{j=1}^J$ denote the ensemble of $J$ particles at $k$-th alternating iteration of the SGS framework. In the likelihood step (Eq. (4)), we aim to sample $\boldsymbol{z}^{(j)}$ from $\pi^{Z|X=\boldsymbol{x}^{(j)}}(\boldsymbol{z}) \propto \exp(-f(\boldsymbol{z}; \boldsymbol{y}) - \frac{1}{2\rho^2}\|\boldsymbol{z} - \boldsymbol{x}^{(j)}\|_2^2)$ for each $j \in \{1, \dots, J\}$. Our starting point is the covariance-preconditioned Langevin dynamics with the large particle limit, which is known to have improved conditioning and convergence under quadratic potentials (Reich & Weissmann, 2021; Garbuno-Inigo et al., 2020a):

$$\mathrm{d}\boldsymbol{z}_t^{(j)} = -C_t \nabla \left( f(\boldsymbol{z}_t^{(j)}; \boldsymbol{y}) + \frac{1}{2\rho^2}\|\boldsymbol{z}_t^{(j)} - \boldsymbol{x}^{(j)}\|_2^2 \right) \mathrm{d}t + \sqrt{2C_t}\mathrm{d}\boldsymbol{w}_t, \tag{6}$$

where $q_t$ is the particle distribution, $\bar{\boldsymbol{z}}_t := \mathbb{E}_{q_t}[\boldsymbol{z}_t]$, and $C_t := \mathbb{E}_{q_t}[(\boldsymbol{z}_t - \bar{\boldsymbol{z}}_t)(\boldsymbol{z}_t - \bar{\boldsymbol{z}}_t)^\top]$. As shown in Lemma 1, Eq. (6) admits $\pi^{Z|X=\boldsymbol{x}^{(j)}}(\boldsymbol{z})$ as its stationary distribution, under the mild assumption that the particle distribution does not collapse to a Dirac measure. For the inverse problem in Eq. (1), we have $f(\boldsymbol{z}_t^{(j)}; \boldsymbol{y}) = \frac{1}{2\sigma_y^2}\|\mathcal{G}(\boldsymbol{z}_t^{(j)}) - \boldsymbol{y}\|_2^2$. Therefore, running Eq. (6) relies on the derivative of the forward model $\mathcal{G}$, which may not be available. To circumvent this, we approximate $\mathcal{G}$ with a linear

surrogate model $\boldsymbol{y} = A_t \boldsymbol{z}_t + \boldsymbol{b}_t$ with the minimal least square error defined by:

$$\min_{A_t, \boldsymbol{b}_t} \mathbb{E}_{q_t} \|\mathcal{G}(\boldsymbol{z}_t) - (A_t \boldsymbol{z}_t + \boldsymbol{b}_t)\|_2^2. \qquad \text{(statistical linearization)}$$

Setting the derivatives w.r.t. $A_t$ and $\boldsymbol{b}_t$ to zero gives the closed-form solution:

$$A_t = \mathbb{E}_{q_t}[(\mathcal{G}(\boldsymbol{z}_t) - \mathbb{E}_{q_t}\mathcal{G}(\boldsymbol{z}_t))(\boldsymbol{z}_t - \bar{\boldsymbol{z}}_t)^\top]C_t^{-1}, \boldsymbol{b}_t = \mathbb{E}_{q_t}\mathcal{G}(\boldsymbol{z}_t) - \mathbb{E}_{q_t}A_t\boldsymbol{z}_t, \qquad (7)$$

where $C_t^{-1}$ is the pseudo-inverse of the covariance matrix. This approach, known as statistical linearization, was first introduced in Booton (1954) and recently used in Kim et al. (2024); Zheng et al. (2025a). Statistical linearization is exact when $\mathcal{G}$ is linear. Let $D\mathcal{G}$ denote the Jacobian of $\mathcal{G}$ and $\tilde{\boldsymbol{z}}_t = \boldsymbol{z}_t - \bar{\boldsymbol{z}}_t$. Replacing $D\mathcal{G}$ with $DA_t = A_t$ in $\nabla f(\boldsymbol{z}_t^{(j)}; \boldsymbol{y})$ yields:

$$\nabla f(\boldsymbol{z}_t^{(j)}; \boldsymbol{y}) = \frac{1}{\sigma_y^2}(D^\top \mathcal{G})(\mathcal{G}(\boldsymbol{z}_t^{(j)}) - \boldsymbol{y}) \approx \frac{1}{\sigma_y^2}C_t^{-1}\mathbb{E}_{q_t}[\tilde{\boldsymbol{z}}_t(\mathcal{G}(\boldsymbol{z}_t) - \mathbb{E}_{q_t}\mathcal{G}(\boldsymbol{z}_t))^\top](\mathcal{G}(\boldsymbol{z}_t^{(j)}) - \boldsymbol{y}). \quad (8)$$

Substituting Eq. (8) into Eq. (6) gives us the derivation of the likelihood step:

$$\mathrm{d}\boldsymbol{z}_t^{(j)} = -\left[\frac{1}{\sigma_y^2}\mathbb{E}_{q_t}[\tilde{\boldsymbol{z}}_t(\mathcal{G}(\boldsymbol{z}_t) - \mathbb{E}_{q_t}\mathcal{G}(\boldsymbol{z}_t))^\top](\mathcal{G}(\boldsymbol{z}_t^{(j)}) - \boldsymbol{y}) + \frac{1}{\rho^2}C_t(\boldsymbol{z}_t^{(j)} - \boldsymbol{x}^{(j)})\right]\mathrm{d}t + \sqrt{2C_t}\mathrm{d}\boldsymbol{w}_t, \quad (9)$$

which does not require explicit computation of $\hat{A}$ and thus avoids computing the matrix pseudo-inverse. Eq. (9) also eliminates the need for derivatives of the forward model, allowing us to run the algorithm with only black-box access to $\mathcal{G}$. The dynamics in Eq. (9) shares a similar structure as that of Ensemble Kalman Sampling (EKS) (Garbuno-Inigo et al., 2020a) and ALDI (Garbuno-Inigo et al., 2020b). However, a key distinction is that each particle in `Blade` has its own target distribution associated with $\boldsymbol{x}^{(j)}$, enabling multi-modal sampling, whereas EKS and ALDI share the same potential function across all the particles.

## 3.2 ENSEMBLE-BASED PRIOR STEP VIA DENOISING DIFFUSION

Let $\mathbf{Z}^{(k)} = \{\boldsymbol{z}^{(j)}\}_{j=1}^J$ denote the ensemble of $J$ particles at $k$-th alternating iteration of the SGS framework. In the prior step (Eq. (5)), we aim to sample $\boldsymbol{x}^{(j)}$ from $\pi^{X|Z=\boldsymbol{z}^{(j)}}(\boldsymbol{x}) \propto \exp(-g(\boldsymbol{x}) - \frac{1}{2\rho^2}\|\boldsymbol{x} - \boldsymbol{z}^{(j)}\|_2^2)$ for each $j \in \{1, \ldots, J\}$. As shown in Coeurdoux et al. (2023); Wu et al. (2024), the prior step can be formulated as denoising diffusion process. Specifically, recall that the forward process gives $p(\boldsymbol{x}_t \mid \boldsymbol{x}_0) = \mathcal{N}(\boldsymbol{x}_t; \boldsymbol{x}_0, \sigma(t)^2\mathbf{I})$ under the EDM (Karras et al., 2022) framework, where $s(t) = 1$. By Bayes' theorem, we have:

$$p(\boldsymbol{x}_0 \mid \boldsymbol{x}_t) \propto p(\boldsymbol{x}_t \mid \boldsymbol{x}_0)p(\boldsymbol{x}_0) \propto \exp\left(-g(\boldsymbol{x}_0) - \frac{1}{2\sigma^2(t)}\|\boldsymbol{x}_0 - \boldsymbol{x}_t\|_2^2\right). \qquad (10)$$

By comparing Eq. (10) with the target distribution $\pi^{X|Z=\boldsymbol{z}^{(j)}}(\boldsymbol{x})$, we can see that if $\rho = \sigma(t)$ and $\boldsymbol{x}_t = \boldsymbol{z}^{(j)}$, sampling from $\pi^{X|Z=\boldsymbol{z}^{(j)}}(\boldsymbol{x})$ is equivalent to sampling from $p(\boldsymbol{x}_0|\boldsymbol{x}_t = \boldsymbol{z}^{(j)})$. Therefore, the prior step of $j$-th particle can be implemented as the standard reverse process of the diffusion model given by Eq. (2) starting from $\boldsymbol{z}^{(j)}$ at time $t^*$ where $t^*$ is chosen so that $\sigma(t^*) = \rho$.

## 3.3 PRACTICAL ALGORITHM

**Practical implementation of likelihood step.** To implement the dynamics in Eq. (9) with a finite-particle system, we introduce two practical variants: `Blade(main)` and `Blade(diag)`. The former is our main algorithm, designed to preserve correct posterior uncertainty, while the latter often provides sharper point estimates. The pseudocode is provided in Algorithm 2 in the appendix.

- `Blade (main)` ensures that the invariant measure of the finite-particle system remains the same as that of Eq. (9). As shown in Nüsken & Reich (2019), the covariance-preconditioned stochastic process requires additional correction term as the diffusion term depends on the evolving particle. For $j$-th particle, we add a correction term to the drift of Eq. (9), yielding:

$$\mathrm{d}\boldsymbol{z}_t^{(j)} = -\left[\frac{1}{\sigma_{\boldsymbol{y}}^2}C_tA_t^\top(\mathcal{G}(\boldsymbol{z}_t^{(j)}) - \boldsymbol{y}) + \frac{1}{\rho^2}C_t(\boldsymbol{z}_t^{(j)} - \boldsymbol{x}^{(j)})\right]\mathrm{d}t + \sqrt{2C_t}\mathrm{d}\boldsymbol{w}_t + \frac{n+1}{J}(\boldsymbol{z}_t^{(j)} - \bar{\boldsymbol{z}}_t)\mathrm{d}t, \quad (11)$$

where $n$ is the dimensionality of $z$, $J$ is the ensemble size, and $\boldsymbol{w}_t \in \mathbb{R}^J$. Lemma 3 verifies that Eq. (11) has an invariant measure that is identical to that of Eq. (9). Intuitively, the correction term $\frac{n+1}{J}(\boldsymbol{z}_t^{(j)} - \bar{\boldsymbol{z}}_t)$ pushes the particles away from each other and vanishes when $J \gg n$. For the computation of $\sqrt{C_t}$, we use the construction proposed in Garbuno-Inigo et al. (2020b) where $\sqrt{C_t} = \frac{1}{\sqrt{J}}(\boldsymbol{z}_t^{(1)} - \bar{\boldsymbol{z}}_t, \ldots, \boldsymbol{z}_t^{(J)} - \bar{\boldsymbol{z}}_t) \in \mathbb{R}^{n \times J}$, which avoids explicit matrix square roots.

- `Blade (diag)` simply approximates the $\sqrt{C_t}$ by the diagonal standard deviation. This simplification introduces a force that pulls the particles into each other, resulting in a smaller spread. It often yields sharper point estimates which can lead to over-confidence.

**Implementation of prior step.** Detailed pseudocode for the prior step can be found in Algorithm 3. Note that in Algorithm 3, $\mathbf{X}_i$ represents the ensemble of particles at $t_i$ step, and the updates for all particles can be computed in parallel. For discretization, we use the Euler method with the step size scheme in Karras et al. (2022). Further implementation details are deferred to Appendix B.1.

**Putting it all together.** We provide pseudocode for the complete sampling algorithm in Algorithm 1. The method operates by iteratively updating an ensemble of particles, alternating between the likelihood and prior steps discussed above. At the same time, the parameter $\rho$ follows an annealing schedule that gradually decreases towards zero. Annealing $\rho$ from large to small implements a smooth path from a smooth posterior with good mixing to the sharp one with improved accuracy. Further details are deferred to Appendix B.

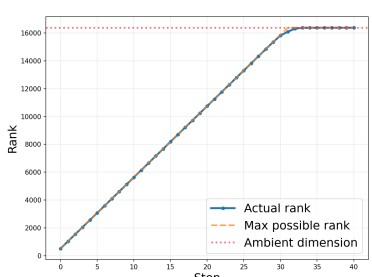

Figure 3: Evolution of the rank of the space spanned by ensemble particles during `Blade` iterations.

*Remark* 1. Like all other ensemble Kalman methods, when the ensemble size is finite, the statistical linearization in a single likelihood step of `Blade` is confined to a subspace spanned by the particles, meaning that it is typically not a full-rank update, which is sometimes problematic in other ensemble Kalman frameworks. However, since `Blade` alternates between likelihood and prior steps, the randomness and nonlinearity of the prior step generally implies that sampling trajectory will explore all dimensions (assuming a full-rank prior). To empirically assess whether Blade suffers from the same limitation, we track the dimensionality of explored space over iterations in the setting where the ensemble size (512) is significantly smaller than the ambient dimension (16384). At iteration $k$, we form the matrix whose columns are the concatenated particles from all iterations up to $k$: $[X^0, Z^0, \ldots, X^{(k)}, Z^{(k)}]$. The accumulated rank is the rank of this matrix. Fig. 3 shows that the actual rank closely tracks the max possible rank and it reaches the ambient dimension within 35 steps, indicating that `Blade` has sufficient posterior exploration without the low-dimensional confinement that limits other ensemble methods.

### 3.4 THEORETICAL ANALYSIS

In this section, we analyze the behavior of `Blade` through the lens of its continuous-time and large particle limit for the ease of understanding. In practice, `Blade` incurs two bias terms: $\epsilon_{\text{model}}$ from the statistical linearization and $\epsilon_{\text{score}}$ from the learned diffusion prior. By extending existing proof techniques from Wu et al. (2024); Vempala & Wibisono (2019); Sun et al. (2024), our analysis quantifies how these errors together with the number of iterations $K$ affects the deviation from the reference process. Technical definitions and notations are collected in Appendix A.1.

**Theorem 1** (Stationary distribution). *Given any $\rho > 0$, consider the oracle split-Gibbs algorithm that alternates between the likelihood step defined in Eq. (6) and the prior step defined in Eq. (2) where each step is implemented perfectly without approximations. If the particle distribution is not a Dirac measure, then $\pi^{XZ}$ is a stationary distribution. Furthermore, if the preconditioner $C_t$ is positive definite, $\pi^{XZ}$ is the unique stationary distribution.*

We defer the proof to Appendix A.2.

*Remark* 2. Theorem 1 shows that, when every sub-step is exact, $\pi^{XZ}$ is the stationary distribution under the standard argument with positive definite preconditioner. Note that Theorem 1 holds for any positive definite preconditioner and the covariance matrix is just one special case. The preconditioner does not change the stationary distribution but rather specifies the geometry of the dynamics (See

Kalman-Wasserstein gradient flow in Garbuno-Inigo et al. (2020a)). This observation will be used in Theorem 2 to compare two processes within a common metric structure.

The next theorem establishes convergence and stability of `Blade` under the two practical approximation errors. The prior score approximation error arising from the diffusion model is defined in Assumption 1, which coincides with the usual bounded score-matching error condition that underpins convergence results (Lee et al., 2022; Chen et al., 2022; Wu et al., 2024). The statistical linearization error $\epsilon_{\text{model}}$ defined in Assumption 2 characterizes the L2 accuracy of the statistical linearization. It is uniformly bounded under the standard regularity assumptions on forward model and uniformly bounded covariance matrix. We also make Assumption 3, which is the weakest condition to bound the weighted Fisher divergence in our analysis. Two common sufficient (but not necessary) scenarios are: (1) $C_t$ is full-rank (2) $\mu_t$ is absolutely continuous with respect to $\tilde{\mu}_t$ and both log densities are continuously differentiable.

**Theorem 2** (Convergence analysis). *Given $\rho > 0$, consider the following two processes that alternate between the likelihood step with horizon $t^\dagger$ and the prior step with horizon $t^*$, where $\sigma(t^*) = \rho$:*

- *The approximate process that implements the likelihood step as in Eq. (9) (with forward model approximation) and the prior step as in Eq. (2) (with diffusion model score approximation). Let $\tilde{\mu}_t$ denote its distribution at time $t$, $C_t$ the associated covariance matrix, $\lambda_t^*$ the smallest non-zero eigenvalue of $C_t$.*

- *The reference process that starts from the stationary distribution $\pi^{XZ}$, implements the likelihood step as Eq. (6) with the preconditioner $C_t$, and the prior step which runs Eq. (2), assuming exact knowledge of both the prior score function and forward model derivative. Let $\mu_t$ denote its distribution at time $t$.*

*Let $T_k = k(t^\dagger + t^*), k = 0, \ldots, K$, $\lambda^* = \inf_{t \in \cup_k [T_k, T_k + t^\dagger]} \lambda_t^*$, and $\delta = \inf_{t \in [0, t^*]} \delta(t)$ where $\delta(t)$ is the diffusion term defined in Eq. (14). We assume both $\lambda^*$ and $\delta$ are strictly positive. We denote by $\epsilon_{\text{score}}$ the score approximation error of the diffusion model defined in Assumption 1, and $\epsilon_{\text{model}}$ the forward model derivative approximation error defined in Assumption 2. Assuming that $D_{\text{KL}}(\pi^X || \mu_0) < +\infty$ and Assumption 3 holds, for $K$ split Gibbs iterations, we have*

$$\frac{1}{T_K} \int_0^{T_K} D_{\text{FI}}(\mu_t || \tilde{\mu}_t) \mathrm{d}t \leq \frac{4}{\min(\lambda^*, \delta)} \left[ \frac{D_{\text{KL}}(\pi^X || \tilde{\mu}_0)}{K(t^\dagger + t^*)} + \frac{t^\dagger \epsilon_{\text{model}} + t^* \epsilon_{\text{score}}}{t^\dagger + t^*} \right] \quad (12)$$

*where $D_{\text{FI}}$ and $D_{\text{KL}}$ are Fisher divergence and KL divergence respectively, defined in Appendix A.1.*

The proof of is deferred to Appendix A.2.

*Remark* 3. Theorem 2 provides two main insights. The first is the convergence to the reference process. The time-average Fisher divergence between the approximate process and reference process decays at an $O(1/K)$ rate up to a weighted sum of approximation errors. Second, the algorithm remains stable under the statistical linearization error and the prior score approximation error and these terms do not vanish with more iterations as expected.

**Interpretation of the reference process.** The reference process stays at the target stationary distribution if the likelihood step is exact regardless of the chosen preconditioner $C_t$. The role of preconditioner $C_t$ here is to specify the geometry with respect to which we measure approximation error as mentioned in Remark 2. Using $C_t$ from the approximate process to construct the reference process does not alter the target stationary distribution; it simply allows us to compare the two dynamics within a common metric structure. Hence, the convergence to the reference process indicates the convergence to the desired stationary distribution.

**Comparison to prior work** While the bound in Eq. 12 is structurally similar to that of `PnPDM` (Wu et al., 2024), our analysis strictly generalizes the prior results to a more realistic and technically challenging regime. First, our method considers a different and more complex likelihood step dynamics with interacting particles and state-dependent preconditioner, whereas Wu et al. (2024) considers the standard Langevin dynamics. Second, our setting is more realistic by considering finite time execution of the dynamics as well as the forward model error. Third, Theorem 2 explicitly shows how the algorithm remains stable under forward model error and how the finite time horizon affect convergence, which are not addressed in the prior work.

## 4 EXPERIMENTS

Inverse problems are ill-posed, so an ensemble of posterior samples with calibrated uncertainty is often more desirable than a single point estimate. We therefore evaluate `Blade` through probabilistic verification of its posterior samples. Sec. 4.1 considers fully controlled settings with analytic ground-truth posteriors, enabling direct distributional checks. Sec. 4.2 turns to a challenging high-dimensional problem based on the Navier-Stokes equation where the ground-truth posterior is unknown but probabilistic verification methods are available. Sec. 4.3 studies the effect of `Blade`'s hyperparameters and its robustness to diffusion prior choice. Sec. D reports results on popular point-estimate benchmarks as complementary evidence of `Blade`'s breadth.

### 4.1 GAUSSIAN AND GAUSSIAN MIXTURE

To enable direct distributional check, we consider two cases where the exact posteriors can be derived: (i) a linear Gaussian posterior and (ii) a two-component non-isotropic Gaussian mixture posterior. For both we derive the analytic posterior and draw "ground-truth" samples, then compare to those produced by different algorithms. We compare `Blade` against the existing derivative-free methods including DPG (Tang et al., 2024), SCG (Huang et al., 2024), EnKG (Zheng et al., 2025a), and EKS (Garbuno-Inigo et al., 2020a). Fig. 1 shows scatter plots of the joint samples alongside marginal density estimates. In the linear Gaussian test (Fig. 1, top row), `Blade` and EKS, as sampling methods, are able to capture the posterior mean and variance. In contrast, optimization-based algorithms like SCG, DPG, and EnKG operate by minimizing a surrogate objective and thus return point estimates, failing to capture posterior spread/uncertainty. In the multimodal non-isotropic Gaussian mixture case (Fig. 1, 2nd row), `Blade` recovers both modes and their relative weights while EKS yields an over-dispersed Gaussian approximation. This is because the particles of EKS share a single potential function while `Blade` assign individual potentials to particles, enabling multi-modal posterior sampling (as mentioned in Sec. 3.1). The detailed setup, derivations of the analytic posteriors, and additional evaluation metrics, are in Appendix C.1, with further quantitative results in Appendix D.

### 4.2 NAVIER-STOKES EQUATION

To test `Blade` in a high-dimensional and challenging setting that is closer to the real-world problems, we consider the problem of recovering the initial vorticity field in the two-dimensional Navier–Stokes equations from partial, noisy observations taken at a later time. We pick this problem because the dynamics is highly nonlinear and mirrors many practical challenges in science and engineering including weather data assimilation (White, 2000), geophysics (Liu & Gurnis, 2008), and fluid reconstruction (Elsinga et al., 2006). More importantly, this setting has standard probabilistic verifications available to test the quality of posterior samples.

**Problem setup.** We take the Navier-Stokes problem formulation in InverseBench (Zheng et al., 2025b) where a non-trivial distribution is considered. The initial vorticity $x^*$ in resolution $128 \times 128$ is evolved forward with a numerical solver, then subsampled and corrupted with Gaussian noise of standard deviation $\sigma_{\text{noise}} = 0, 1, 2$. The observation $y$ thus constitute a partial, noisy snapshot of the flow field. We use the publicly released dataset and pretrained diffusion prior from InverseBench. All the experiments are conducted on single GH200 GPU. Details are in Appendix C.2.

**Baselines.** We compare our algorithm against two classes of methods. The first class is the methods that only requires diffusion prior trained on unpaired data including DPG (Tang et al., 2024), SCG (Huang et al., 2024), EnKG (Zheng et al., 2025a), EKI (Iglesias et al., 2013), EKS (Garbuno-Inigo et al., 2020a) (initialized from diffusion prior), EKS W/ DP (EKS with diffusion prior), localized EKS w/ DP (Reich & Weissmann, 2021; Wagner et al., 2022), and FKD (Singhal et al., 2025; Zhao et al., 2025). The second class is provided as reference points, which requires additional training on paired data, including conditional diffusion model (CDM) and an end-to-end U-Net. The conditional diffusion learns the posterior distribution through conditional score matching. The U-Net directly learns to predict ground truth from the observation. For each noise regime we retrain both the conditional diffusion model and the U-Net from scratch, using the same training configuration. The details are in Appendix C.2.3.

**Evaluation metrics.** For comprehensive evaluation, we consider three different metrics in the literature (Zheng et al., 2025b; Rasp et al., 2024) to assess the performance from both probabilistic and deterministic perspectives: continuous ranked probability score (CRPS), spread-skill ratio (SSR),

Table 1: Comparison on the Navier-Stokes inverse problem. The primary probabilistic metrics are CRPS and SSR. Rel L2 error (relative L2 error) is deterministic, included as a complementary metric. $-$ indicates either that probabilistic metrics are inapplicable (deterministic method) or that it is too costly to generate enough samples from the algorithm for reliable calculation of probabilistic metrics. CDM-CA: conditional diffusion model with cross attention. CDM-Cat: conditional diffusion model with channel concatenation.

| | $\sigma_{\text{noise}} = 0$ | | | $\sigma_{\text{noise}} = 1.0$ | | | $\sigma_{\text{noise}} = 2.0$ | | |
|---|---|---|---|---|---|---|---|---|---|
| | CRPS↓ | SSR→1 | Rel L2 error↓ | CRPS↓ | SSR→1 | Rel L2 error↓ | CRPS↓ | SSR→1 | Rel L2 error↓ |
| **Paired data** | | | | | | | | | |
| CDM-CA | 2.900 | 0.983 | 1.362 | 2.872 | 1.059 | 1.409 | 2.993 | 1.087 | 1.542 |
| CDM-Cat | 1.413 | 0.896 | 0.653 | 1.805 | 0.979 | 0.873 | 2.211 | 0.974 | 1.043 |
| U-Net | — | — | 0.585 | — | — | 0.702 | — | — | 0.709 |
| **Unpaired data** | | | | | | | | | |
| EKI | 2.303 | 0.012 | 0.577 | 2.350 | 0.118 | 0.586 | 2.700 | 0.011 | 0.673 |
| EKS + DM | 1.900 | 0.181 | 0.539 | 2.088 | 0.218 | 0.606 | 2.255 | 0.280 | 0.685 |
| EKS w/ DP | 1.280 | 0.061 | 0.336 | 1.723 | 0.085 | 0.455 | 2.094 | 0.080 | 0.547 |
| Localized EKS w/ DP | 1.643 | 0.056 | 0.428 | 1.887 | 0.081 | 0.495 | 2.057 | 0.089 | 0.542 |
| FKD | 1.604 | 0.002 | 0.399 | 1.416 | 0.050 | 0.368 | 1.810 | 0.012 | 0.455 |
| DPG | — | — | 0.325 | — | — | 0.408 | — | — | 0.466 |
| SCG | — | — | 0.961 | — | — | 0.928 | — | — | 0.966 |
| EnKG | 0.395 | 0.164 | 0.120 | 0.651 | 0.154 | 0.191 | 1.032 | 0.144 | 0.294 |
| Blade (diag) | 0.276 | 0.086 | **0.080** | 0.542 | 0.177 | **0.162** | 0.758 | 0.129 | **0.217** |
| Blade (main) | **0.216** | **0.955** | 0.110 | **0.453** | **0.950** | 0.229 | **0.608** | **0.949** | 0.306 |

and relative L2 error (Rel L2 error). CRPS is a proper scoring rule that rewards both sharp predictive distribution and well-calibrated predictions whereas SSR diagnoses calibration only. An SSR near one is desirable but must be interpreted in conjunction with the other error metrics. Formal definitions and implementation details of these metrics are in Appendix C.2.2.

**Results.** Table 1 and Fig. 1 (c) summarizes performance under three observation noise levels. Blade (main) offers the best calibrated ensemble predictions: its CRPS is the best among all, and its SSR remains close to one. The other competing methods are too confident (SSR $< 0.2$) and their predictions do not represent the true uncertainty. The CRPS of CDM has a very high CRPS despite SSR near one, which means that it produces overly diffuse distribution with large errors. Blade(diag) delivers the best point estimates with the lowest relative L2 but is under-dispersive with larger CRPS and small SSR compared to Blade (main), confirming our theoretical insights in Sec.3.3. Overall, Blade (main) achieves both accurate recovery and reliable uncertainty calibration. We provide qualitative comparison in Fig. 1 (b) (full comparison with rank histograms are in Fig. 9). The runtime comparison is reported in Fig. 14. As shown, Blade runs slower than EKS but faster than the other derivative-free baselines. Moreover, its inherently parallel design allows for straightforward scaling, similar to common data parallelism.

## 4.3 ABLATION STUDIES

**Test-time scaling.** We vary the number of split-Gibbs alternations $K$ and track the performance of Blade across different observation noise levels. As shown in Fig. 4 (b), the predictive performance and calibration of Blade both improve rapidly up to $K = 20$. Beyond roughly 20 iterations the performance gains start to plateau, so additional computation yields diminishing returns. This general empirical trend aligns with the $O(1/K)$ decaying effect in Eq. (12).

**Impact of different hyperparameters.** We perform comprehensive controlled experiments to understand the impact of different hyperparameters and design choices. Fig. 4 (a) sweeps three main hyperparameters of Blade. Formal definitions of hyperparameters, detailed discussion, and additional results are in Appendix B. In a nutshell, we observe that the discretization step scale $\gamma$ and the minimum coupling strength $\rho_{\min}$ both have a broad plateau where both accuracy and calibration remain near-optimal, simplifying practical tuning. $\tilde{\sigma}_{\boldsymbol{y}}$ is the most important factor that affects the performance. Increasing $\tilde{\sigma}_y$ widens the ensemble as expected. A sweet spot region is around 0.15.

**Data scaling.** We evaluate how the performance of Blade depends on the pre-trained diffusion prior. We retrain the diffusion prior with progressively larger data subsets (ranging from $10 \times 2^7$ to

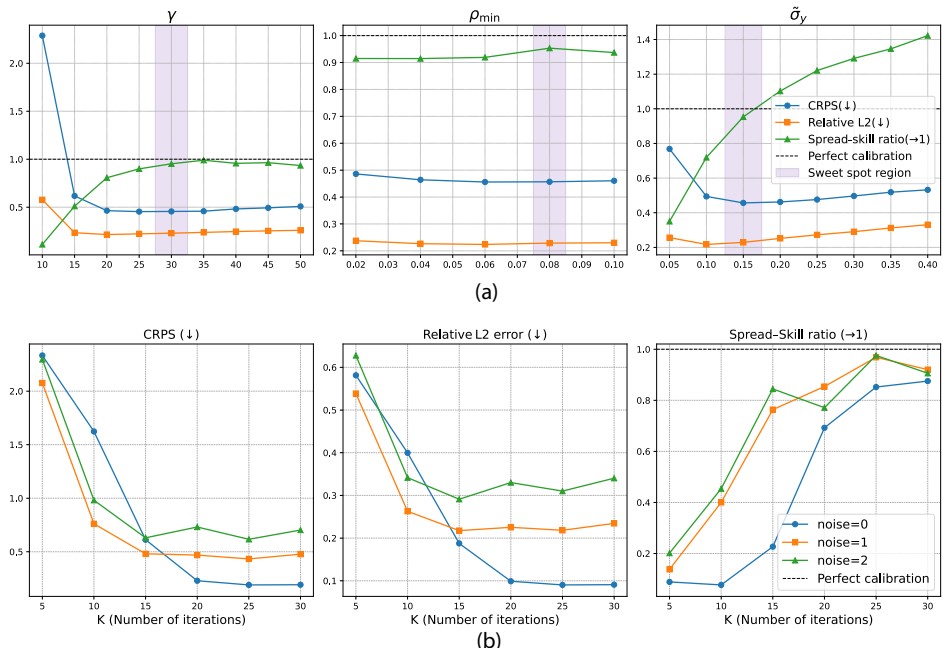

Figure 4: (a): Effect of different hyperparameters. $\gamma$: discretization step scale; $\rho_{\min}$: the minimum coupling strength. $\tilde{\sigma}_{\boldsymbol{y}}$: the likelihood–spread factor. (b): Test-time scaling of `Blade` across different measurement noise levels. With more split Gibbs iterations, Blade not only becomes more accurate but also provides a more reliable assessment of its uncertainty.

$10 \times 2^{11}$). We compare the resulting performance to that of the end-to-end U-Net and CDM, both retrained for each subset and each noise level. As shown in Fig. 10, `Blade` surpasses the baselines by a significant margin. In contrast, the U-Net improves only modestly with data size. These results indicate that `Blade` is sample-efficient, can efficiently exploit extra prior data when available yet maintaining decent performance even in low-data regime.

**Additional ablation studies.** We conduct additional ablation studies of `Blade` in Appendix B, including the design choice of the annealing schedule for the coupling strength, the impact of initialization, the ensemble size, and the effect of resample strategy.

## 5 CONCLUSION

In this paper, we introduced `Blade`, a derivative-free, ensemble-based Bayesian inversion algorithm for inverse problems without direct derivative information. Built on the split Gibbs framework, `Blade` alternates between a derivative-free likelihood step and a denoising diffusion prior step. Experiments demonstrate its accuracy and reliable uncertainty quantification, and our theory provides explicit error bounds accounting for both statistical linearization and the learned score function.

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

## A  THEORY

### A.1  NOTATION

We denote by $h(t)$ the drift coefficient in Eq. (2) and $\delta(t)$ the diffusion coefficient:

$$h(t) := -\left(2\dot{\sigma}(t)\sigma(t) + \beta(t)\right) \tag{13}$$

$$\delta(t) := \sqrt{2\dot{\sigma}(t)\sigma(t)} + \sqrt{2\beta(t)}. \tag{14}$$

The Kullback–Leibler (KL) divergence between two distributions $\mu$ and $\tilde{\mu}$ is $D_{\mathrm{KL}}$ defined by

$$D_{\mathrm{KL}}(\mu||\tilde{\mu}) = \int \mu \log \frac{\mu}{\tilde{\mu}} = \mathbb{E}_{\mu} \log \frac{\mu}{\tilde{\mu}}.$$

The Fisher divergence between two distributions $\mu$ and $\tilde{\mu}$ is $D_{\mathrm{FI}}$ defined by

$$D_{\mathrm{FI}}(\mu||\tilde{\mu}) = \int \mu \|\nabla \log \frac{\mu}{\tilde{\mu}}\|_2^2 = \mathbb{E}_{\mu} \left\| \nabla \log \frac{\mu}{\tilde{\mu}} \right\|_2^2.$$

For a positive semi-definite matrix $B \in \mathbb{R}^{n \times n}$, we denote by $\|\cdot\|_B$ the weighted norm defined by

$$\|u\|_B^2 = u^\top B u, \tag{15}$$

where $u \in \mathbb{R}^n$. For $\boldsymbol{x} \in \mathbb{R}^n$, the divergence of a matrix $T(\boldsymbol{x}) \in \mathbb{R}^{n \times n}$ is the vector field:

$$(\nabla_{\boldsymbol{x}} \cdot T)_i = \sum_{j=1}^n \frac{\partial T_{ij}}{\boldsymbol{x}_j}. \tag{16}$$

### A.2  PROOFS

**Assumption 1.** *The average score approximation error of the diffusion model $s_\theta$ is bounded,*

$$\epsilon_{\mathrm{score}} = \sup_{k=0,\ldots,K-1} \left\{ \frac{1}{t^*} \int_{T_k+t^\dagger}^{T_{k+1}} \frac{h(t)^2}{\delta(t)^2} \mathbb{E}_{\mu_t} \|\nabla_{\boldsymbol{x}_t} \log p\left(\boldsymbol{x}_t; \sigma(t)\right) - s_\theta(\boldsymbol{x}_t, t)\|_2^2 \mathrm{d}t \right\} < +\infty, \tag{17}$$

*where $h(t)$ is defined in Eq. (13) and $\delta(t)$ is defined in Eq. (14).*

**Assumption 2.** *The average derivative approximation error of the linear surrogate model $A_t$ is bounded,*

$$\epsilon_{\mathrm{model}} = \sup_{k=0,\ldots,K-1} \left\{ \frac{1}{t^\dagger} \int_{T_k}^{T_k+t^\dagger} \mathbb{E}_{\mu_t} \left\| \nabla f(\boldsymbol{z}_t; \boldsymbol{y}) - \frac{1}{\sigma_y^2} A_t^\top (\mathcal{G}(\boldsymbol{z}_t^{(j)}) - \boldsymbol{y}) \right\|_{C_t}^2 \mathrm{d}t \right\} < +\infty, \tag{18}$$

*where $\|\cdot\|_{C_t}$ is the weighted norm defined in Eq. (15).*

**Assumption 3.** *The Radon–Nikodym derivative $\frac{\mathrm{d}\mu_t}{\mathrm{d}\tilde{\mu}_t}$ is constant along the null space of $C_t$ almost surely, where $C_t$ is the covariance matrix of $\tilde{\mu}_t$.*

$$\frac{\mathrm{d}\mu_t}{\mathrm{d}\tilde{\mu}_t}(\boldsymbol{x}) = \frac{\mathrm{d}\mu_t}{\mathrm{d}\tilde{\mu}_t}(\boldsymbol{x} + \boldsymbol{v}), \forall \boldsymbol{v} \in \mathrm{Ker}(C_t).$$

**Lemma 1** (Stationary distribution of the likelihood step)**.** *Assume the particle distribution is not a Dirac measure, the dynamics of Eq. (6) admits $\pi^{Z|X=\boldsymbol{x}^{(j)}}(\boldsymbol{z}) \propto \exp(-f(\boldsymbol{z}; \boldsymbol{y}) - \frac{1}{2\rho^2}\|\boldsymbol{z} - \boldsymbol{x}^{(j)}\|_2^2)$ as a stationary distribution. Further, if the covariance matrix is positive definite, the stationary distribution is unique.*

*Proof.* This result has been proved in various forms in the literature (Ma et al., 2015; Garbuno-Inigo et al., 2020a), we provide a simple proof of our use case for ease of understanding. Suppose $\mu_t(\boldsymbol{z})$ is the probability density of $\boldsymbol{z}$ at time $t$. For the ease of notation, we ignore the particle index $j$ in $\boldsymbol{z}_t$. Let $\Phi(\boldsymbol{z}) = f(\boldsymbol{z}; \boldsymbol{y}) + \frac{1}{2\rho^2}\|\boldsymbol{z} - \boldsymbol{x}^{(j)}\|_2^2$. The corresponding Fokker-Planck equation for Eq. (6) reads

$$\frac{\partial \mu_t}{\partial t} = \nabla \cdot (\mu_t C_t \nabla \Phi(\boldsymbol{z})) + \nabla \cdot (C_t \nabla \mu_t),$$

which can be rewritten as

$$\frac{\partial \mu_t}{\partial t} = \nabla \cdot (\mu_t C_t (\nabla \Phi(\boldsymbol{z}) + \nabla \log \mu_t)). \tag{19}$$

Let $\mu_\infty$ denote the stationary distribution of Eq. (19). We have

$$0 = \nabla \cdot (\mu_t C_t (\nabla \Phi(\boldsymbol{z}) + \nabla \log \mu_\infty)).$$

If the particle distribution is not Dirac, $C_t \neq 0$ due to Lemma 2.1 in Garbuno-Inigo et al. (2020a). Therefore,

$$\nabla \Phi(\boldsymbol{z}) + \nabla \log \mu_\infty = c,$$

where $c$ is a constant. Integrating both sides gives

$$\mu_\infty(\boldsymbol{z}) \propto \exp(-\Phi(\boldsymbol{z})) = \exp(-f(\boldsymbol{z}; \boldsymbol{y}) - \frac{1}{2\rho^2} \|\boldsymbol{z} - \boldsymbol{x}^{(j)}\|_2^2),$$

showing that $\pi^{Z|X=\boldsymbol{x}^{(j)}}(\boldsymbol{z})$ is a stationary distribution of the dynamics of Eq. (6). Further, if $C_t$ is positive definite, it ensures the irreducibility and strong Feller property, and the stationary distribution is unique (Roberts & Tweedie, 1996; Ma et al., 2015). □

**Theorem 1** (Stationary distribution). *Given any $\rho > 0$, consider the oracle split-Gibbs algorithm that alternates between the likelihood step defined in Eq. (6) and the prior step defined in Eq. (2) where each step is implemented perfectly without approximations. If the particle distribution is not a Dirac measure, then $\pi^{XZ}$ is a stationary distribution. Furthermore, if the preconditioner $C_t$ is positive definite, $\pi^{XZ}$ is the unique stationary distribution.*

*Proof.* We prove this by directly verifying the invariance property, i.e., if the samples $(\boldsymbol{x}, \boldsymbol{z})$ are from the joint distribution $\pi^{XZ}$, then after one iteration of the algorithm, the new samples $(\boldsymbol{x}', \boldsymbol{z}')$ stay in the same distribution $\pi^{XZ}$. By Lemma 1, $\pi^{Z|X=\boldsymbol{x}}$ is a stationary distribution Eq. (6). Therefore, after the oracle likelihood step, a stationary joint density of $(\boldsymbol{x}, \boldsymbol{z}')$ is given by

$$p(\boldsymbol{x}, \boldsymbol{z}') = \int \pi^{XZ}(\boldsymbol{x}, \boldsymbol{z}) \pi^{Z|X=\boldsymbol{x}}(\boldsymbol{z}') \mathrm{d}\boldsymbol{z} = \pi^X(\boldsymbol{x}) \pi^{Z|X=\boldsymbol{x}}(\boldsymbol{z}'),$$

where $\pi^X$ is the marginal distribution. As shown in Eq. (10), after sampling $\boldsymbol{x}'$ given $\boldsymbol{z}'$ according to the prior step, the joint density of $(\boldsymbol{x}', \boldsymbol{z}')$ becomes

$$\begin{aligned}
p(\boldsymbol{x}', \boldsymbol{z}') &= \int p(\boldsymbol{x}, \boldsymbol{z}') \pi^{X|Z=\boldsymbol{z}'}(\boldsymbol{x}') \mathrm{d}\boldsymbol{x} \\
&= \int \pi^X(\boldsymbol{x}) \pi^{Z|X=\boldsymbol{x}}(\boldsymbol{z}') \pi^{X|Z=\boldsymbol{z}'}(\boldsymbol{x}') \mathrm{d}\boldsymbol{x} \\
&= \pi^Z(\boldsymbol{z}') \pi^{X|Z=\boldsymbol{z}'}(\boldsymbol{x}') \\
&= \pi^{XZ}(\boldsymbol{x}', \boldsymbol{z}'),
\end{aligned}$$

showing that the distribution of $(\boldsymbol{x}', \boldsymbol{z}')$ remains $\pi^{XZ}$ after one round of updates. Therefore, $\pi^{XZ}$ is a stationary distribution. Furthermore, by Lemma 1, if the particle covariance remains positive definite, the stationary distribution of the likelihood step is unique. Consequently, it follows that $\pi^{XZ}$ is the unique stationary distribution. □

**Lemma 2.** *Given the following pair of stochastic processes*

$$\mathrm{d}\boldsymbol{x}_t = b(\boldsymbol{x}_t, t)\mathrm{d}t + H(t)\mathrm{d}\boldsymbol{w}_t, \tag{20}$$

$$\mathrm{d}\tilde{\boldsymbol{x}}_t = \tilde{b}(\tilde{\boldsymbol{x}}_t, t)\mathrm{d}t + H(t)\mathrm{d}\boldsymbol{w}_t, \tag{21}$$

*where $b, \tilde{b} : \mathbb{R}^n \times \mathbb{R}^+ \to \mathbb{R}^n$ are the drift terms, $H : \mathbb{R}^+ \to \mathbb{R}^n \times \mathbb{R}^n$ is the diffusion term, $\boldsymbol{w}_t$ is the standard Wiener process. Let $\mu_t$ (respectively $\tilde{\mu}_t$) be the law of $\boldsymbol{x}_t$ (respectively $\tilde{\boldsymbol{x}}_t$), $C(t) := H(t)H(t)^\top$, and $\lambda_t^*$ be the smallest non-zero eigenvalue of $C(t)$. Assuming that $b_t - \tilde{b}_t \in$ Range($C(t)$) and Assumption 3 holds, we have*

$$\frac{\partial}{\partial t} D_{\mathrm{KL}}(\mu_t || \tilde{\mu}_t) \leq -\frac{\lambda_t^*}{4} D_{\mathrm{FI}}(\mu_t || \tilde{\mu}_t) + \mathbb{E}_{\mu_t} \left\| b_t - \tilde{b}_t \right\|_{C(t)^\dagger}^2, \tag{22}$$

*where $C(t)^\dagger$ is the pseudo-inverse of $C(t)$.*

*Proof.* Since the diffusion terms only depend on $t$ and $C(t) = H(t)H(t)^\top$, the Fokker-Planck equations of Eq. (20) and Eq. (21) read

$$\frac{\partial}{\partial t}\mu_t = \nabla \cdot \left[ \left( \frac{1}{2}C(t)\nabla \log \mu_t - b_t \right) \mu_t \right], \tag{23}$$

$$\frac{\partial}{\partial t}\tilde{\mu}_t = \nabla \cdot \left[ \left( \frac{1}{2}C(t)\nabla \log \tilde{\mu}_t - \tilde{b}_t \right) \tilde{\mu}_t \right]. \tag{24}$$

Let $r_t := \frac{\mu_t}{\tilde{\mu}_t}$ and $\phi(r_t) := r_t \log r_t$ (so $\phi'(r_t) = \frac{\mathrm{d}}{\mathrm{d}r_t}\phi(r_t) = \log r_t + 1$). Differentiating the KL divergence gives

$$\frac{\partial}{\partial t}D_{\mathrm{KL}}(\mu_t||\tilde{\mu}_t) = \frac{\partial}{\partial t}\int \phi(r_t)\tilde{\mu}_t$$

$$= \int \left( \phi(r_t)\frac{\partial \tilde{\mu}_t}{\partial t} + \phi'(r_t)\frac{\partial r_t}{\partial t}\tilde{\mu}_t \right)$$

$$= \int \left( \phi(r_t)\frac{\partial \tilde{\mu}_t}{\partial t} + \phi'(r_t)\frac{\partial \mu_t}{\partial t} - \phi'(r_t)r_t\frac{\partial \tilde{\mu}_t}{\partial t} \right)$$

$$= \int \left( (\log r_t + 1)\frac{\partial \mu_t}{\partial t} - r_t\frac{\partial \tilde{\mu}_t}{\partial t} \right), \tag{25}$$

where the last step uses the fact that $\phi(r_t) - r_t\phi'(r_t) = -r_t$. Plugging Eq. (23) and Eq. (24) into Eq. (25) and applying integration by parts further gives

$$\frac{\partial}{\partial t}D_{\mathrm{KL}}(\mu_t||\tilde{\mu}_t)$$

$$= \int (\log r_t + 1)\nabla \cdot \left[ \left( \frac{1}{2}C(t)\nabla \log \mu_t - b_t \right) \mu_t \right] - \int r_t \nabla \cdot \left[ \left( \frac{1}{2}C(t)\nabla \log \tilde{\mu}_t - \tilde{b}_t \right) \tilde{\mu}_t \right]$$

$$= -\int \left\langle \nabla \log r_t, \frac{1}{2}C(t)\nabla \log \mu_t - b_t \right\rangle \mu_t + \int \left\langle \nabla r_t, \frac{1}{2}C(t)\nabla \log \tilde{\mu}_t - \tilde{b}_t \right\rangle \tilde{\mu}_t$$

$$= -\int \left\langle \nabla \log r_t, \frac{1}{2}C(t)\nabla \log \mu_t - b_t \right\rangle \mu_t + \int \left\langle \nabla \log r_t, \frac{1}{2}C(t)\nabla \log \tilde{\mu}_t - \tilde{b}_t \right\rangle \mu_t$$

$$= -\int \left\langle \nabla \log r_t, \frac{1}{2}C(t)\left( \nabla \log \mu_t - \nabla \log \tilde{\mu}_t \right) \right\rangle \mu_t + \int \left\langle \nabla \log r_t, b_t - \tilde{b}_t \right\rangle \mu_t$$

$$= -\frac{1}{2}\int \left\langle \nabla \log r_t, C(t)\nabla \log r_t \right\rangle \mu_t + \int \left\langle \nabla \log r_t, b_t - \tilde{b}_t \right\rangle \mu_t. \tag{26}$$

The weighted Young's inequality states that, for any $u, v \in \mathbb{R}^n$, when $v \in \mathrm{Range}(C)$, we have

$$\langle u, v \rangle \leq \frac{1}{4}\langle u, Cu \rangle + \langle v, C^\dagger v \rangle,$$

where $C^\dagger$ is the pseudo-inverse. By Assumption 3, Eq. (26) can be bounded as follows

$$-\frac{1}{2}\int \langle \nabla \log r_t, C(t)\nabla \log r_t \rangle \mu_t + \int \left\langle \nabla \log r_t, b_t - \tilde{b}_t \right\rangle \mu_t$$

$$\leq -\frac{1}{4}\int \langle \nabla \log r_t, C(t)\nabla \log r_t \rangle \mu_t + \int \left\langle b_t - \tilde{b}_t, C(t)^\dagger(b_t - \tilde{b}_t) \right\rangle \mu_t$$

$$\leq -\frac{\lambda_t^*}{4}D_{\mathrm{FI}}(\mu_t||\tilde{\mu}_t) + \mathbb{E}_{\mu_t}\left\| b_t - \tilde{b}_t \right\|_{C(t)^\dagger}^2 \tag{27}$$

where $\lambda_t^*$ is the smallest non-zero eigenvalue of $C(t)$. $\qquad\square$

*Remark* 4. This is a generalization of Lemma A.4 in Wu et al. (2024) to the general matrix-valued diffusion term. Intuitively, the condition that $b_t - \tilde{b}_t$ belongs to the range of $C(t)$ means that the two drift terms may only differ along the directions that are actually driven by noise. In the context of our proof below, this is always satisfied because the drift terms are either preconditioned with $C(t)$ or $C(t)$ is full-rank.

**Theorem 2** (Convergence analysis). *Given $\rho > 0$, consider the following two processes that alternate between the likelihood step with horizon $t^\dagger$ and the prior step with horizon $t^*$, where $\sigma(t^*) = \rho$:*

- *The approximate process that implements the likelihood step as in Eq. (9) (with forward model approximation) and the prior step as in Eq. (2) (with diffusion model score approximation). Let $\tilde{\mu}_t$ denote its distribution at time $t$, $C_t$ the associated covariance matrix, $\lambda_t^*$ the smallest non-zero eigenvalue of $C_t$.*

- *The reference process that starts from the stationary distribution $\pi^{XZ}$, implements the likelihood step as Eq. (6) with the preconditioner $C_t$, and the prior step which runs Eq. (2), assuming exact knowledge of both the prior score function and forward model derivative. Let $\mu_t$ denote its distribution at time $t$.*

*Let $T_k = k(t^\dagger + t^*), k = 0, \ldots, K$, $\lambda^* = \inf_{t \in \cup_k [T_k, T_k + t^\dagger]} \lambda_t^*$, and $\delta = \inf_{t \in [0,t^*]} \delta(t)$ where $\delta(t)$ is the diffusion term defined in Eq. (14). We assume both $\lambda^*$ and $\delta$ are strictly positive. We denote by $\epsilon_{\text{score}}$ the score approximation error of the diffusion model defined in Assumption 1, and $\epsilon_{\text{model}}$ the forward model derivative approximation error defined in Assumption 2. Assuming that $D_{\text{KL}}(\pi^X \| \mu_0) < +\infty$ and Assumption 3 holds, for $K$ split Gibbs iterations, we have*

$$\frac{1}{T_K} \int_0^{T_K} D_{\text{FI}}(\mu_t \| \tilde{\mu}_t) \mathrm{d}t \leq \frac{4}{\min(\lambda^*, \delta)} \left[ \frac{D_{\text{KL}}(\pi^X \| \tilde{\mu}_0)}{K(t^\dagger + t^*)} + \frac{t^\dagger \epsilon_{\text{model}} + t^* \epsilon_{\text{score}}}{t^\dagger + t^*} \right] \quad (12)$$

*where $D_{\text{FI}}$ and $D_{\text{KL}}$ are Fisher divergence and KL divergence respectively, defined in Appendix A.1.*

*Proof.* For $t \in [T_k, T_k + t^\dagger], k = 0, \ldots, K - 1$, we apply Lemma 2 to the likelihood step with

$$b(\boldsymbol{z}_t, t) := -C_t \nabla f(\boldsymbol{z}_t; \boldsymbol{y}) - \frac{1}{\rho^2} C_t (\boldsymbol{z}_t - \boldsymbol{x}^{(j)})$$

$$\tilde{b}(\boldsymbol{z}_t, t) := -C_t \frac{1}{\sigma_y^2} A_t^\top (\mathcal{G}(\boldsymbol{z}_t) - \boldsymbol{y}) - \frac{1}{\rho^2} C_t (\boldsymbol{z}_t - \boldsymbol{x}^{(j)})$$

$$H(t) = \sqrt{C_t},$$

where $A_t = \mathbb{E}_{\tilde{\mu}_t} [(\mathcal{G}(\boldsymbol{z}_t) - \mathbb{E}_{q_t} \mathcal{G}(\boldsymbol{z}_t)) \boldsymbol{z}_t^\top] C_t^{-1}$ as defined in Eq. (7). Note that the condition $b - \tilde{b} \in \text{Range}(C_t)$ is satisfied as both drift terms are preconditioned with $C_t$. Thus, by Assumption 3, we have

$$\frac{\partial}{\partial t} D_{\text{KL}}(\mu_t \| \tilde{\mu}_t) \leq -\frac{\lambda_t^*}{4} D_{\text{FI}}(\mu_t \| \tilde{\mu}_t) + \mathbb{E}_{\mu_t} \left\langle (b_t - \tilde{b}_t), C_t^\dagger (b_t - \tilde{b}_t) \right\rangle$$

$$\leq -\frac{\lambda^*}{4} D_{\text{FI}}(\mu_t \| \tilde{\mu}_t) + \mathbb{E}_{\mu_t} \left\| \nabla f(\boldsymbol{z}_t; \boldsymbol{y}) - \frac{1}{\sigma_y^2} A_t^\top (\mathcal{G}(\boldsymbol{z}_t) - \boldsymbol{y}) \right\|_{C_t}^2$$

where $\lambda_t^*$ is the smallest non-zero eigenvalue of $C_t$ and $\lambda^* := \inf \lambda_t^*$. By Assumption 2, integrating both sides over $[T_k, T_k + t^\dagger]$ gives

$$D_{\text{KL}}(\mu_{T_k + t^\dagger} \| \tilde{\mu}_{T_k + t^\dagger}) - D_{\text{KL}}(\mu_{T_k} \| \tilde{\mu}_{T_k})$$

$$\leq -\frac{\lambda_t^*}{4} \int_{T_k}^{T_k + t^\dagger} D_{\text{FI}}(\mu_t \| \tilde{\mu}_t) \mathrm{d}t + \int_{T_k}^{T_k + t^\dagger} \mathbb{E}_{\mu_t} \left\| \nabla f(\boldsymbol{z}_t; \boldsymbol{y}) - \frac{1}{\sigma_y^2} A_t^\top (\mathcal{G}(\boldsymbol{z}_t) - \boldsymbol{y}) \right\|_{C_t}^2 \mathrm{d}t$$

$$\leq -\frac{\lambda_t^*}{4} \int_{T_k}^{T_k + t^\dagger} D_{\text{FI}}(\mu_t \| \tilde{\mu}_t) \mathrm{d}t + t^\dagger \epsilon_{\text{model}}, \quad (28)$$

where $\epsilon_{\text{model}}$ is defined in Eq. (18). For $t \in [T_k + t^\dagger, T_{k+1}], k = 0, \ldots, K - 1$, we apply Lemma 2 to the prior step (2) with

$$b(\boldsymbol{x}_t, t) := h(t) \nabla_{\boldsymbol{x}_t} \log p(\boldsymbol{x}_t; \sigma(t))$$

$$\tilde{b}(\boldsymbol{z}_t, t) := h(t) s_\theta(\boldsymbol{x}_t, t)$$

$$H(t) := \delta(t) \mathbf{I},$$

where $h(t)$ is the drift coefficient defined in Eq. (13), $\delta(t)$ is the diffusion coefficient defined in Eq. (14), $s_\theta$ is the pre-trained diffusion model with score approximation error $\epsilon_{\text{score}}$. Note that $H(t)H(t)^\top$ is full-rank so that the condition of Lemma 2 is satisfied. Therefore, we have

$$\frac{\partial}{\partial t} D_{\text{KL}}(\mu_t || \tilde{\mu}_t) \leq -\frac{\delta(t)^2}{4} D_{\text{FI}}(\mu_t || \tilde{\mu}_t) + \frac{h(t)^2}{\delta(t)^2} \mathbb{E}_{\mu_t} \|\nabla_{\boldsymbol{x}_t} \log p(\boldsymbol{x}_t; \sigma(t)) - s_\theta(\boldsymbol{x}_t, t)\|_2^2$$

$$\leq -\frac{\delta}{4} D_{\text{FI}}(\mu_t || \tilde{\mu}_t) + \frac{h(t)^2}{\delta(t)^2} \|\nabla_{\boldsymbol{x}_t} \log p(\boldsymbol{x}_t; \sigma(t)) - s_\theta(\boldsymbol{x}_t, t)\|_2^2,$$

where $\delta := \inf_{t \in [0, t^*]} \delta(t)^2$. Integrating both sides over $[T_k + t^\dagger, T_{k+1}]$ and applying Assumption 1 gives

$$D_{\text{KL}}(\mu_{T_{k+1}} || \tilde{\mu}_{T_{k+1}}) - D_{\text{KL}}(\mu_{T_k + t^\dagger} || \tilde{\mu}_{T_k + t^\dagger})$$

$$\leq -\frac{\delta}{4} \int_{T_k + t^\dagger}^{T_{k+1}} D_{\text{FI}}(\mu_t || \tilde{\mu}_t) \mathrm{d}t + \int_{T_k + t^\dagger}^{T_{k+1}} \frac{h(t)^2}{\delta(t)^2} \mathbb{E}_{\mu_t} \|\nabla_{\boldsymbol{x}_t} \log p(\boldsymbol{x}_t; \sigma(t)) - s_\theta(\boldsymbol{x}_t, t)\|_2^2 \mathrm{d}t$$

$$\leq -\frac{\delta}{4} \int_{T_k + t^\dagger}^{T_{k+1}} D_{\text{FI}}(\mu_t || \tilde{\mu}_t) \mathrm{d}t + t^* \epsilon_{\text{score}}, \tag{29}$$

where $\epsilon_{\text{score}}$ is defined in Eq. (17). Summing up both sides of Eq. (28) and Eq. (29) for $k = 0, \ldots, K - 1$ gives

$$D_{\text{KL}}(\mu_{T_K} || \tilde{\mu}_{T_K}) - D_{\text{KL}}(\mu_0 || \tilde{\mu}_0) \leq -\frac{\min(\lambda^*, \delta)}{4} \int_0^{T_K} D_{\text{FI}}(\mu_t || \tilde{\mu}_t) \mathrm{d}t + K(t^\dagger \epsilon_{\text{model}} + t^* \epsilon_{\text{score}}).$$

Rearranging the terms gives

$$\frac{1}{T_K} \int_0^{T_K} D_{\text{FI}}(\mu_t || \tilde{\mu}_t) \mathrm{d}t$$

$$\leq \frac{4}{T_K \min(\lambda^*, \delta)} \left( D_{\text{KL}}(\mu_0 || \tilde{\mu}_0) - D_{\text{KL}}(\mu_{T_K} || \tilde{\mu}_{T_K}) \right) + \frac{4}{\min(\lambda^*, \delta)(t^\dagger + t^*)} (\epsilon_{\text{model}} + \epsilon_{\text{score}})$$

$$\leq \frac{4}{\min(\lambda^*, \delta)} \left[ \frac{D_{\text{KL}}(\mu_0 || \tilde{\mu}_0)}{K(t^\dagger + t^*)} + \frac{t^\dagger \epsilon_{\text{model}} + t^* \epsilon_{\text{score}}}{t^\dagger + t^*} \right].$$

Note that $\mu_0 = \pi^X$ and we conclude the proof. $\qquad \square$

**Lemma 3.** *Let $\pi(\boldsymbol{z}; \boldsymbol{x}^{(j)})$ denote the invariant measure associated with the potential $\Phi(\boldsymbol{z}; \boldsymbol{x}^{(j)})$ where $\nabla_{\boldsymbol{z}} \Phi(\boldsymbol{z}; \boldsymbol{x}^{(j)}) = \left[ \frac{1}{\sigma_{\boldsymbol{y}}^2} A_t^\top (\mathcal{G}(\boldsymbol{z}) - \boldsymbol{y}) + \frac{1}{\rho^2} (\boldsymbol{z} - \boldsymbol{x}^{(j)}) \right]$. Then $\pi(\boldsymbol{z}; \boldsymbol{x}^{(j)})$ is an invariant measure of the finite-particle system in Eq. (11) as well as its large particle limit in Eq. (9).*

*Proof.* In the large particle limit, the covariance $C_t$ does not depend on any specific particle but depends on the particle distribution only. Therefore, the Fokker-Plank equation of Eq. (9) reads:

$$\frac{\partial}{\partial t} p_t = \nabla \cdot \left( p_t C_t \nabla \Phi(\boldsymbol{z}_t^{(j)}; \boldsymbol{x}^{(j)}) \right) + C_t \nabla^2 p_t$$

$$= \nabla \cdot \left( p_t C_t \left( \nabla \Phi(\boldsymbol{z}_t^{(j)}; \boldsymbol{x}^{(j)}) + \nabla \log p_t \right) \right)$$

where $p_t$ is the probability density at time $t$. We can see that $\pi(\boldsymbol{z}; \boldsymbol{x}^{(j)})$ is an invariant measure by setting both sides to zero. In the finite-particle system, the covariance $C_t = \frac{1}{J} \sum_{j=1}^J (\boldsymbol{z}_t^{(j)} - \bar{\boldsymbol{z}}_t)(\boldsymbol{z}_t^{(j)} - \bar{\boldsymbol{z}}_t)^\top$, which depends on the current state $\boldsymbol{z}_t^{(j)}$. Therefore, the Fokker-Plank equation of

the finite-particle dynamics in Eq. (11) is

$$
\begin{aligned}
\frac{\partial}{\partial t} p_t =& \nabla \cdot \left[ p_t \left( C_t \nabla \Phi(z_t^{(j)}; x^{(j)}) - \frac{n+1}{J}(z_t^{(j)} - \bar{z}_t) \right) \right] + \nabla \cdot (\nabla \cdot (p_t C_t)) \\
=& \nabla \cdot \left[ p_t \left( C_t \nabla \Phi(z_t^{(j)}; x^{(j)}) - \frac{n+1}{J}(z_t^{(j)} - \bar{z}_t) \right) \right] + \nabla \cdot (C_t \nabla p_t + p_t \nabla \cdot C_t) \\
=& \nabla \cdot \left[ p_t C_t \left( \nabla \Phi(z_t^{(j)}; x^{(j)}) + \nabla \log p_t \right) \right] - \nabla \cdot \left( \frac{n+1}{J}(z_t^{(j)} - \bar{z}_t) \right) + \nabla \cdot (p_t \nabla \cdot C_t) \\
=& \nabla \cdot \left[ p_t C_t \left( \nabla \Phi(z_t^{(j)}; x^{(j)}) + \nabla \log p_t \right) \right] - \nabla \cdot \left( \frac{n+1}{J}(z_t^{(j)} - \bar{z}_t) \right) \\
& + \nabla \cdot \left( p_t \nabla_{z_t^{(j)}} \cdot \frac{1}{J} \sum_{i=1}^{J}(z_t^{(i)} - \bar{z}_t)(z_t^{(i)} - \bar{z}_t)^\top \right) \\
=& \nabla \cdot \left[ p_t C_t \left( \nabla \Phi(z_t^{(j)}; x^{(j)}) + \nabla \log p_t \right) \right] - \nabla \cdot \left( p_t \frac{n+1}{J}(z_t^{(j)} - \bar{z}_t) \right) \\
& + \nabla \cdot \left( p_t \frac{1}{J}(n+1)(z_t^{(j)} - \bar{z}_t) \right) \\
=& \nabla \cdot \left[ p_t C_t \left( \nabla \Phi(z_t^{(j)}; x^{(j)}) + \nabla \log p_t \right) \right]
\end{aligned}
$$

where the divergence of a matrix is defined in Eq. (16), and we use the following properties:

$$
\begin{aligned}
\nabla_{z_t^{(j)}} \cdot \left( z_t^{(j)} z_t^{(j)\top} \right) &= (n+1) z_t^{(j)}, \\
\nabla_{z_t^{(j)}} \cdot \left( z_t^{(j)} z_t^{(i)\top} \right) &= z_t^{(i)}, \\
\nabla_{z_t^{(j)}} \cdot \left( z_t^{(i)} z_t^{(j)\top} \right) &= n z_t^{(i)}, \\
\nabla_{z_t^{(j)}} \cdot \left( \bar{z}_t \bar{z}_t^\top \right) &= \frac{n+1}{J} \bar{z}_t,
\end{aligned}
$$

where $i \neq j$. By taking both sides to zero, we have that $\pi(z; x^{(j)})$ is an invariant measure of Eq. (11). □

*Remark* 5. This proof is largely adapted from Nüsken & Reich (2019); Garbuno-Inigo et al. (2020b) which applies to more general scenarios. We tailor and simplify the proof for our use case for the ease of understanding.

## B  PRACTICAL IMPLEMENTATION

In this section, we detail the practical implementation of each component of the proposed method, `Blade`. We also present ablation studies for each design choice to elucidate their individual effects. All the ablation studies are conducted on a small subset of the Navier-Stokes inverse problem's test set.

### B.1  LIKELIHOOD STEP

**Initialization**  As Theorem 2 indicates, the initialization of `Blade` is quite flexible, provided the initial distribution maintains a finite KL divergence from the target distribution. Our empirical evaluation considered two initialization strategies: Gaussian and diffusion prior (DM) initialization. For the Navier-Stokes inverse problem, as shown in Fig. 12, the empirical performance difference between two initializations is not substantial. In contrast, for image restoration tasks, we observed that Gaussian initialization gets better results. This is because the target distribution being more closely approximated by a Gaussian due to the large initial coupling strength. Conversely, DM initialization tends to produce a natural image distribution, which is known to often exhibit an unbounded KL divergence from the Gaussian (Arjovsky & Bottou, 2017). Therefore, we use Gaussian initialization for the image restoration tasks.

---

**Algorithm 2** Ensemble-based likelihood sampling step

---

**Require:** forward model $\mathcal{G}$, observation $\boldsymbol{y}$, effective observation noise $\tilde{\sigma}_y$, coupling strength $\rho$, number of discretization steps $N$, step size scale $\gamma$, initial ensemble $\mathbf{X} = \{\boldsymbol{x}^{(j)}\}_{j=1}^{J}$, mode (`main` or `diag`).

1: $\mathbf{Z}_0 \leftarrow \mathbf{X}$

2: **for** $i \in \{0, \ldots, N-1\}$ **do**

3:      $\epsilon_i \sim \mathcal{N}(0, \mathbf{I})$

4:      $\mathrm{d}_1^{(j)} \leftarrow -\frac{1}{\tilde{\sigma}_y^2} \frac{1}{J} \sum_{k=1}^{J} \langle \mathcal{G}(\boldsymbol{z}_i^{(k)}) - \bar{\mathcal{G}}, \mathcal{G}(\boldsymbol{z}_i^{(j)}) - \boldsymbol{y} \rangle (\boldsymbol{z}_i^{(k)} - \bar{\boldsymbol{z}}_i), j = 1, \ldots, J$

5:      **if** mode is `main` **then**

6:         $\mathrm{d}_2^{(j)} \leftarrow -\frac{1}{\rho^2} C_i(\boldsymbol{x}^{(j)} - \boldsymbol{z}_i^{(j)}) + \frac{n+1}{J}(\boldsymbol{z}_i^{(j)} - \bar{\boldsymbol{z}}_i), j = 1, \ldots, J$

7:         $\sqrt{C_i} := \frac{1}{\sqrt{J}} \left( \boldsymbol{z}_i^{(1)} - \bar{\boldsymbol{z}}_i, \ldots, \boldsymbol{z}_i^{(J)} - \bar{\boldsymbol{z}}_i \right)$

8:      **else**

9:         $\mathrm{d}_2^{(j)} \leftarrow -\frac{1}{\rho^2} C_i(\boldsymbol{x}^{(j)} - \boldsymbol{z}_i^{(j)}), j = 1, \ldots, J$

10:         $\left( \sqrt{C_i} \right)_k \leftarrow \mathrm{std}\left( (\boldsymbol{z}_i)_k \right), k = 1, \ldots, n$

11:      **end if**

12:      $\eta \leftarrow \gamma / \|\mathrm{d}_1 + \mathrm{d}_2\|_2^2$

13:      $\boldsymbol{z}_{i+1}^{(j)} \leftarrow \boldsymbol{z}_i^{(j)} + (\mathrm{d}_1^{(j)} + \mathrm{d}_2^{(j)})\eta + \sqrt{2C_i\eta}\epsilon_i, j = 1, \ldots, J$

14: **end for**

15: **return** $\mathbf{Z}_N$

---

**Algorithm 3** Ensemble prior sampling step

---

**Require:** Diffusion model $s_\theta$, coupling strength $\rho$, number of discretization steps $N$, initial ensemble $\mathbf{Z} = \{\boldsymbol{z}^{(j)}\}_{j=1}^{J}$, $\sigma(t) = t$, $s(t) = 1$, discretization time steps $t_{i \in \{0, \cdots, N\}}$

1: $\mathbf{X}_0 \leftarrow \mathbf{Z}$

2: $i^* \leftarrow \min\{i \geq 0 \mid \sigma(t_i) \leq \rho\}$

3: **for** $i \in \{i^*, \ldots, N-1\}$ **do**

4:      $\lambda \leftarrow 2$ **if** *SDE* **else** 1

5:      $\mathrm{d}_i \leftarrow -\lambda t_i s_\theta(\mathbf{X}_i, \sigma(t_i))$

6:      $\mathbf{X}_{i+1} \leftarrow \mathbf{X}_i + (t_{i+1} - t_i)\mathrm{d}_i$

7:      **if** $i \neq N-1$ **and** *SDE* **then**

8:         $\epsilon_i \sim \mathcal{N}(0, \mathbf{I})$

9:         $\mathbf{X}_{i+1} \leftarrow \mathbf{X}_{i+1} + \sqrt{2t_i(t_i - t_{i+1})}\epsilon_i$

10:      **end if**

11: **end for**

12: **return** $\mathbf{X}_N$

---

**Discretization**   We discretize the SDE in Eq. (11) using the standard Euler method with an adaptive step size defined as

$$\text{Step size} = \gamma \cdot \frac{1}{\|\text{drift}\|_2^2},$$

where $\gamma$ is the hyperparameter that controls the scale of the step size, $\text{drift}$ is the drift term of the SDE in Eq. (11). This adaptive step size is effective across all our experiments. Further design of adaptive step sizes could potentially reduce discretization error with fewer steps. Incorporating the ideas from modern deep learning optimizers for this purpose would be an interesting direction for future work.

**Resample**   During the likelihood step, we employ a resampling strategy to ensure that the particles are at the correct noise level $\rho$. Resampling is a commonly used method that has been shown to help improve the performance of algorithms such as DAPS Zhang et al. (2025), DiffPIR Zhu et al. (2023), and ReSample Song et al. (2023). Specifically, we define the following resampling strategy:

$$\boldsymbol{z}_{\text{resample}}^{(j)} = \boldsymbol{z}^{(j)} + \rho' \epsilon,$$

where $\epsilon \sim \mathcal{N}(0, \mathbf{I})$, $\rho' = \max(0, \rho - \frac{\text{Tr}(C_t)}{n})$, and $n$ is the dimension of the variable $\boldsymbol{z}$. Intuitively, we approximate the current noise level in $\boldsymbol{z}$ and add a corresponding amount of noise to bring $\boldsymbol{z}_{\text{resample}}^{(j)}$ to noise level $\rho$. A key distinction from the prior work is that our $\rho'$ is estimated from the ensemble while the existing methods need to tune it as part of hyperparameters. We present the ablation on the effect of resampling strategy in Fig. 13. As shown, the results with and without resampling are almost the same. In our main experiments, we apply resampling strategy since it introduces minimal additional computation cost and yields slightly better results.

**Effective observation noise**   In practice, we observe that weighting the likelihood with a smaller $\sigma_y$ yields better performance. We denote this adjusted value as the effective observation noise, $\tilde{\sigma}_y$. Using an effective noise smaller than $\sigma_y$ potentially compensates for the smoothing effect introduced by statistical linearization. In practice, we treat $\tilde{\sigma}_y$ as a hyperparameter and tune it so that spread-skill ratio is close to 1. The ablation study present in Fig. 6 demonstrates the effect of this hyperparameter. As anticipated, larger $\tilde{\sigma}_y$ results in an ensemble prediction with greater uncertainty.

## B.2   PRIOR STEP

The prior step is implemented as a denoising diffusion process, with its pseudocode detailed in Algorithm 3. We set $\sigma(t) = t$ for simplicity and employ the Euler ODE sampler for faster sampling. We discretize the denoising diffusion process with the standard Euler method. Following Karras et al. (2022), we use the following step size:

$$t_i = \left( t_{\max}^{1/7} + \frac{i}{N-1}(t_{\min}^{1/7} - t_{\max}^{1/7}) \right)^7, i = 0, \ldots, N-1.$$

## B.3   ANNEALING SCHEDULE

In our experiments, we explored three different types of annealing schedules for the coupling strength $\rho$: linear, EDM, and concave. Given the number of iterations $K$, the maximum value $\rho_{\max}$, and minimum value $\rho_{\min}$, the linear decay schedule reduces $\rho_k$ as

$$\rho_k = \rho_{\max} + \frac{k}{K-1}(\rho_{\min} - \rho_{\max}), k = 0, \ldots, K-1.$$

Inspired by the discretization scheme proposed by Karras et al. (2022), we consider the following schedule, which we refer to as EDM schedule:

$$\rho_k = \left( \rho_{\max}^{1/4} + \frac{k}{K-1}(\rho_{\min}^{1/4} - \rho_{\max}^{1/4}) \right)^4, k = 0, \ldots, K-1.$$

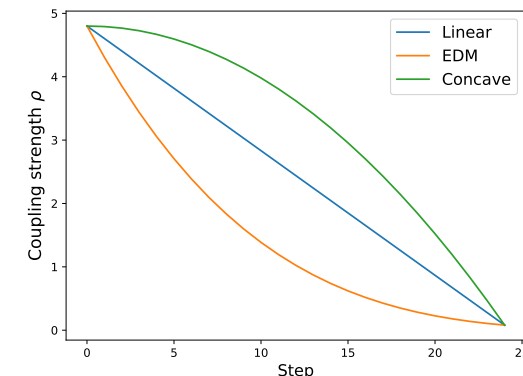

Figure 5: Illustration of the three annealing schedules. Each curve visualizes how the coupling strength $\rho_k$ evolves over iterations.

Table 2: Hyperparameter choices of Blade for the main experiments in Table 1.

| Inverse problem | Mode | $\gamma$ | $\tilde{\sigma}_y$ | $\rho_{\max}$ | $\rho_{\min}$ | $\rho$ schedule | $K$ | $N_{\text{likelihood}}$ | $J$ | Init |
|---|---|---|---|---|---|---|---|---|---|---|
| $\sigma_{\text{noise}} = 0$ | `main` | 20 | 0.03 | 4.8 | 0.08 | `linear` | 25 | 50 | 512 | DM |
| | `diag` | 30 | 0.001 | 4.8 | 0.06 | `concave` | 25 | 50 | 512 | DM |
| $\sigma_{\text{noise}} = 1$ | `main` | 30 | 0.17 | 4.8 | 0.08 | `linear` | 25 | 50 | 512 | DM |
| | `diag` | 35 | 0.04 | 4.8 | 0.08 | `concave` | 25 | 50 | 512 | DM |
| $\sigma_{\text{noise}} = 2$ | `main` | 30 | 0.3 | 4.8 | 0.08 | `linear` | 25 | 50 | 512 | DM |
| | `diag` | 30 | 0.25 | 4.8 | 0.08 | `concave` | 25 | 50 | 512 | DM |

The quadratic concave schedule is designed to decrease slowly at first and accelerate later, defined as a concave transformation of normalized line:

$$\rho_k = \rho_{\min} + (\rho_{\max} - \rho_{\min}) \cdot \left( 1 - \frac{k^2}{(K-1)^2} \right), k = 0, \ldots, K-1.$$

Fig. 5 illustrates the behavior of the three annealing schedules described above. The EDM schedule exhibits a steep initial drop, the linear schedule decays uniformly, and the concave schedule maintains a higher coupling strength initially and then decreases rapidly. We study the difference between these schedules for `Blade (main)` in Fig. 8. All three produce reasonable performance, but simple linear schedule offers the most consistent balance of accuracy and calibration across noise levels. Our default is therefore linear.

### B.4 ENSEMBLE SIZE

We examine the impact of ensemble size on `Blade` in Fig. 11, using the remaining hyperparameters from Table 2. The results show that performance consistently improves with larger ensembles, but the gains plateau around a size of 512. Consequently, we set the ensemble size to 512 for all subsequent experiments.

## C EXPERIMENT DETAILS

### C.1 GAUSSIAN AND GAUSSIAN MIXTURE

#### C.1.1 PROBLEM SETUP

We consider the general linear inverse problem given by

$$\boldsymbol{y} = H\boldsymbol{x} + \epsilon, \tag{30}$$

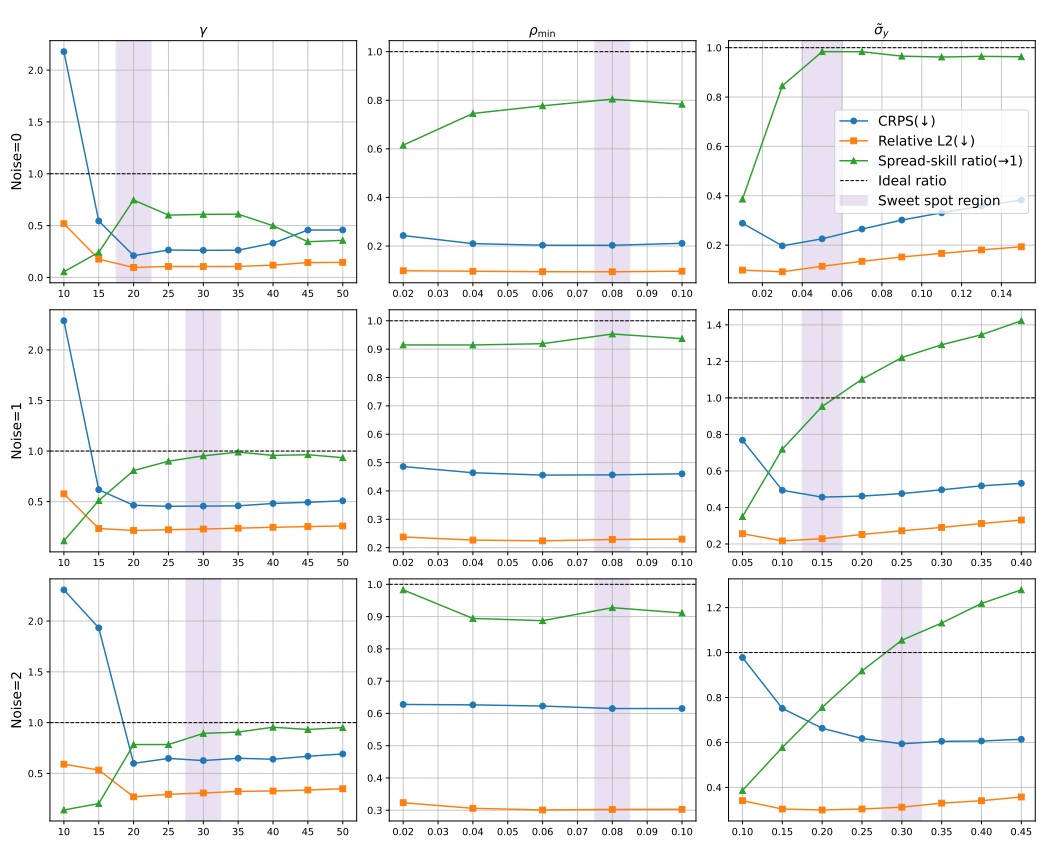

Figure 6: Effect of different hyperparameters of the `Blade` (main) across different measurement noise levels.

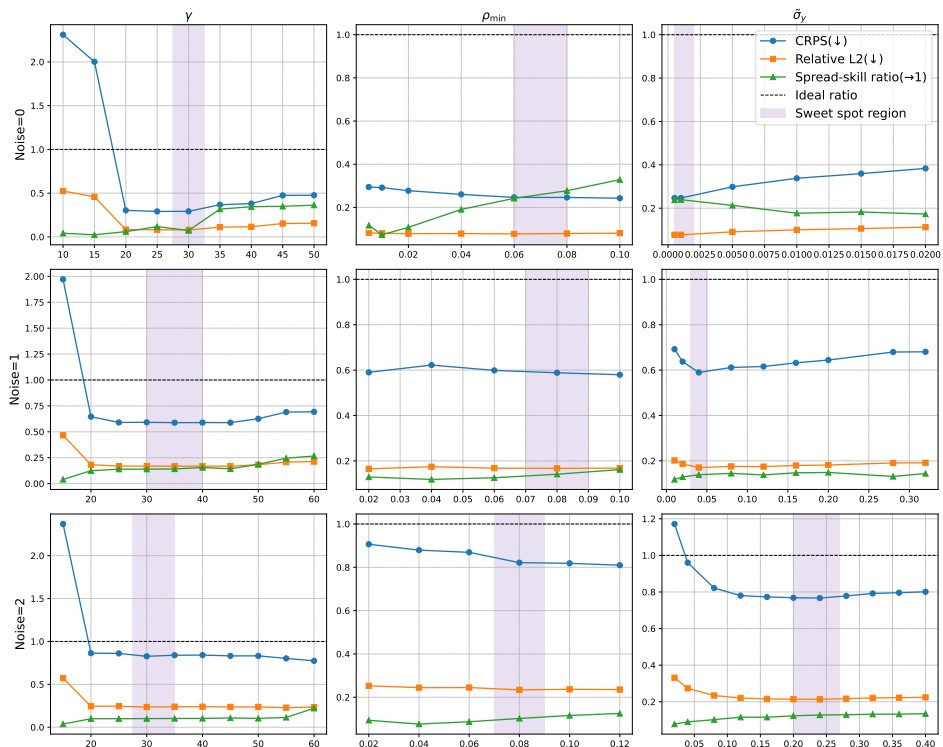

Figure 7: Effect of different hyperparameters of the `Blade` (diag) across different measurement noise levels.

where $\boldsymbol{x} \in \mathbb{R}^n, \boldsymbol{y} \in \mathbb{R}^m, H \in \mathbb{R}^{m \times n}, \epsilon \sim \mathcal{N}(0, \Sigma_\epsilon)$. Given the measurement $\boldsymbol{y}$, we aim to sample from the posterior distribution $p(\boldsymbol{x}|\boldsymbol{y})$. We consider and analyze the case where the prior distribution of $\boldsymbol{x}$ is a mixture of Gaussians given by

$$p(\boldsymbol{x}) = \sum_{i=1}^{K} \gamma_i \mathcal{N}(m_i, \Sigma_i), \sum_{i=1}^{K} \gamma_i = 1, \tag{31}$$

where the mean $m_i \in \mathbb{R}^n$ and the covariance matrix $\Sigma_i \in \mathbb{R}^{n \times n}$. When $K = 1$, the prior degenerates to a Gaussian.

**Linear-Gaussian** In this setting, $K = 1$. For all experiments, we randomly generate $m_1$ and choose $\Sigma_i = 25\mathbf{I}$. We also randomly generate the linear operator $H$. We set $m = 1$ and vary $n$.

**Linear Gaussian mixture** We consider and analyze the case where the prior distribution of $\boldsymbol{x}$ is a mixture of Gaussians given by

$$p(\boldsymbol{x}) = \sum_{i=1}^{K} \gamma_i \mathcal{N}(m_i, \Sigma_i), \sum_{i=1}^{K} \gamma_i = 1, \tag{32}$$

where the mean $m_i \in \mathbb{R}^n$ and the covariance matrix $\Sigma_i \in \mathbb{R}^{n \times n}$.

In our experiments, we set the prior to be a mixture of four Gaussians where the variance of each Gaussian is $2\mathbf{I}$ and the means are $(16i, 16j)$ for $(i, j) \in \{0, 1\}^2$. We set $m = 1, n = 2, \sigma_y^2 = 1.5$. The linear forward model $H$ and observed data $\boldsymbol{y}$ are both randomly generated from Gaussian.

**Ground truth posterior** By linearity, the distribution of $\boldsymbol{y}$ is also a Gaussian mixture given by

$$p(\boldsymbol{y}) = \sum_{i=1}^{K} \gamma_i \mathcal{N}(Hm_i, H\Sigma_i H^\top + \Sigma_\epsilon), \sum_{i=1}^{K} \gamma_i = 1. \tag{33}$$

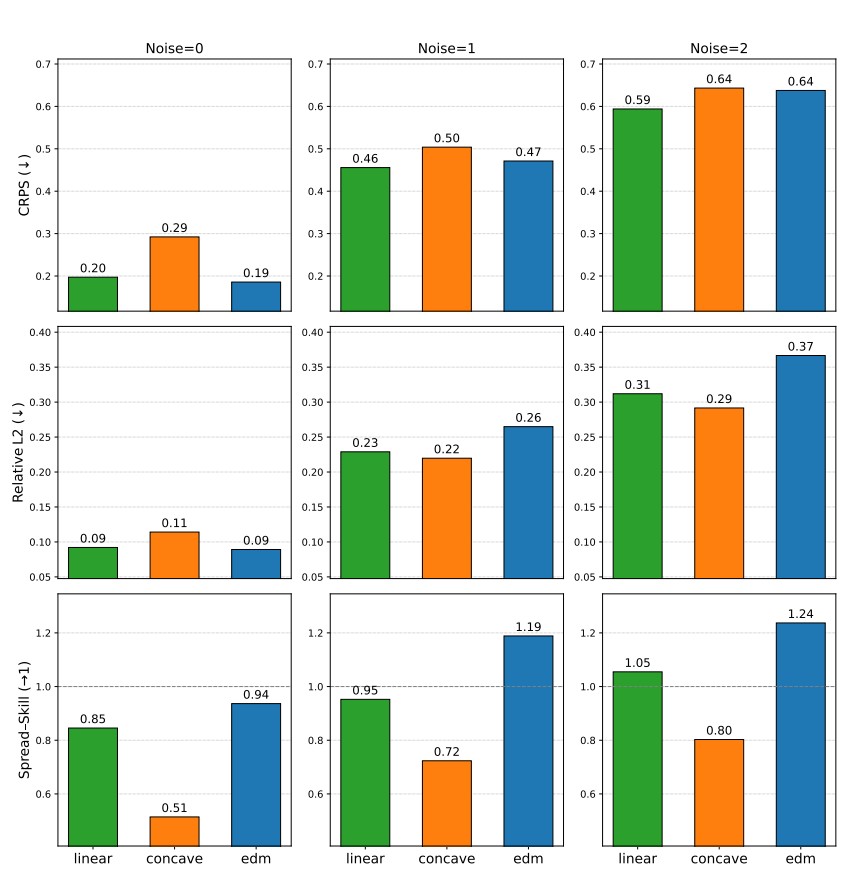

Figure 8: Ablation study on the effect of annealing schedules.

Using Bayes theorem, the posterior distribution is given by

$$p(\boldsymbol{x}|\boldsymbol{y}) = \frac{p(\boldsymbol{y}|\boldsymbol{x})p(\boldsymbol{x})}{p(\boldsymbol{y})}. \tag{34}$$

The likelihood $p(\boldsymbol{y}|\boldsymbol{x})$ reads

$$p(\boldsymbol{y}|\boldsymbol{x}) = \mathcal{N}(\boldsymbol{y}; H\boldsymbol{x}, \Sigma_\epsilon). \tag{35}$$

Therefore,

$$p(\boldsymbol{x}|\boldsymbol{y}) = \frac{\sum_{i=1}^{K} \gamma_i \mathcal{N}(\boldsymbol{x}; m_i, \Sigma_i)\mathcal{N}(\boldsymbol{y}; H\boldsymbol{x}, \Sigma_\epsilon)}{\sum_{i=1}^{K} \gamma_i \mathcal{N}(\boldsymbol{y}; m_i, H\Sigma_i H^\top + \Sigma_\epsilon)}, \tag{36}$$

which can be written as the exponential of a quadratic in $\boldsymbol{x}$. Therefore, the posterior distribution is also a mixture of Gaussians,

$$p(\boldsymbol{x}|\boldsymbol{y}) = \sum_{i=1}^{K} \omega_i \mathcal{N}(\boldsymbol{x}; \hat{m}_i, C_i), \tag{37}$$

where the posterior mean $\hat{m}_i$ and covariance $C_i$ are given by

$$\hat{m}_i = \left(H^\top \Sigma_\epsilon^{-1} H + \Sigma_i^{-1}\right)^{-1} \left(H^\top \Sigma_\epsilon^{-1} \boldsymbol{y} + \Sigma_i^{-1} m_i\right), \tag{38}$$

$$C_i = \left(H^\top \Sigma_\epsilon^{-1} H + \Sigma_i^{-1}\right)^{-1}, \tag{39}$$

and the weight of each mode is given by

$$\omega_j = \frac{\gamma_j \mathcal{N}(\boldsymbol{y}; Hm_j, H\Sigma_j H^\top + \Sigma_\epsilon)}{\sum_{i=1}^{K} \gamma_i \mathcal{N}(\boldsymbol{y}; Hm_i, H\Sigma_i H^\top + \Sigma_\epsilon)}, j = 1, \ldots, K. \tag{40}$$

### C.1.2 EVALUATION METRICS

**KL divergence**  To measure the KL divergence between the distribution of generated samples and the ground truth posterior distribution, we first compute the empirical mean and covariance of the samples. The KL divergence between the $d$-dimensional generated sample distribution $\mathcal{N}(\mu, \Sigma)$ and Gaussian posterior $\mathcal{N}(\mu^*, \Sigma^*)$ is given by

$$D_{\mathrm{KL}}\left(\mathcal{N}(\mu, \Sigma) \,\|\, \mathcal{N}(\mu^*, \Sigma^*)\right) = \frac{1}{2}\left[\log\left(\frac{|\Sigma^*|}{|\Sigma|}\right) - d + \mathrm{tr}\left((\Sigma^*)^{-1}\Sigma\right) + (\mu^* - \mu)^\top (\Sigma^*)^{-1}(\mu^* - \mu)\right].$$

The KL divergence helps quantify the error in both mean and covariance (spread) of the generated samples.

**W2 distance**  We compute the Sliced Wasserstein Distance between the generated samples and samples from the ground truth posterior:

$$\mathrm{SWD}_p(P, Q) \approx \left(\frac{1}{L}\sum_{\ell=1}^{L} W_p^p\left(\langle P, \theta_\ell\rangle, \langle Q, \theta_\ell\rangle\right)\right)^{1/p},$$

where $P$ is a set of generated samples, $Q$ is the set of samples from the ground truth posterior, $\langle P, \theta_\ell\rangle$ and $\langle Q, \theta_\ell\rangle$ denote the empirical 1D distributions formed by projecting the samples onto direction $\theta_\ell$, and $W_p$ is the 1D Wasserstein metric of order $p$. We use the Python Optimal Transport library to compute this metric.

### C.2 NAVIER-STOKES EQUATION

### C.2.1 PROBLEM SETUP

Following the experimental setup in InverseBench (Zheng et al., 2025b), we consider the 2-d Navier-Stokes equation for a viscous, incompressible fluid in vorticity form on a torus,

$$\begin{aligned}
\partial_t \boldsymbol{w}(\boldsymbol{x}, t) + \boldsymbol{u}(\boldsymbol{x}, t) \cdot \nabla \boldsymbol{w}(\boldsymbol{x}, t) &= \nu \Delta \boldsymbol{w}(\boldsymbol{x}, t) + f(\boldsymbol{x}), & \boldsymbol{x} \in (0, 2\pi)^2, t \in (0, T] \\
\nabla \cdot \boldsymbol{u}(\boldsymbol{x}, t) &= 0, & \boldsymbol{x} \in (0, 2\pi)^2, t \in [0, T] \\
\boldsymbol{w}(\boldsymbol{x}, 0) &= \boldsymbol{w}_0(\boldsymbol{x}), & \boldsymbol{x} \in (0, 2\pi)^2
\end{aligned} \tag{41}$$

where $\boldsymbol{u}$ is the velocity field, $\boldsymbol{w} = \nabla \times \boldsymbol{u}$ is the vorticity, $\boldsymbol{w}_0$ is the initial vorticity, $\nu \in \mathbb{R}_+$ is the viscosity coefficient, and $f$ is the forcing function. The solution operator $\mathcal{F}$ is defined as the operator mapping the vorticity from the initial vorticity to the vorticity at time $T$. $\mathcal{F} : \boldsymbol{w}_0 \rightarrow \boldsymbol{w}_T$. Numerically, it is realized as a pseudo-spectral solver (He & Sun, 2007). This Navier-Stokes equation is a standard benchmark problem widely used in the literature (Iglesias et al., 2013; Li et al., 2020; Takamoto et al., 2022). The forward model in our inverse problem is given by

$$\boldsymbol{y} = PL(\mathcal{F}(\boldsymbol{w}_0)) + \epsilon, \tag{42}$$

where $L$ is the discretization operator and $P$ is the sampling operator.

### C.2.2 EVALUATION METRICS

We adopt the following three standard metrics to evaluate the results from different perspectives.

**Relative L2 error**   Suppose $\boldsymbol{x}^*$ is the ground truth function and $\boldsymbol{x}$ is the predicted function. The relative L2 error measures the error $\boldsymbol{x} - \boldsymbol{x}^*$ relative to the norm of the ground truth:

$$\text{Rel L2 error} = \frac{\|\boldsymbol{x} - \boldsymbol{x}^*\|_2}{\|\boldsymbol{x}^*\|_2}.$$

**Continuous Ranked Probability Score (CRPS)**   The CRPS (Gneiting & Raftery, 2007) is a standard probabilistic metric to assess the quality of the entire predicted distribution for inverse problems, which is defined as

$$\text{CRPS} = \mathbb{E}|\boldsymbol{x} - \boldsymbol{x}^*| - \frac{1}{2}\mathbb{E}|\boldsymbol{x} - \boldsymbol{x}'|,$$

where $\boldsymbol{x}, \boldsymbol{x}'$ are independent random predictions and $\boldsymbol{x}^*$ is the single observed ground truth. Intuitively, it measures the distance between a predicted distribution and the single observed ground truth $\boldsymbol{x}^*$ that actually occurred. It is minimized when the ensemble prediction is drawn from the same distribution as the ground truth, i.e., $\boldsymbol{x}^{(j)} \sim p(\boldsymbol{x}^*)$ for all $j$. We consider the multi-dimensional version of CRPS defined in Rasp et al. (2024). For an ensemble prediction $\{\boldsymbol{x}^{(j)}\}_{j=1}^J$ where $\boldsymbol{x}^{(j)} \in \mathbb{R}^n$, the CRPS for the single ground truth $\boldsymbol{x}^*$ is given by

$$\text{CRPS} = \frac{1}{n}\sum_{i=1}^n \left( \frac{1}{J}\sum_{j=1}^J |\boldsymbol{x}^{(j)}(i) - \boldsymbol{x}^*(i)| - \frac{1}{2J(J-1)}\sum_{j=1}^J\sum_{k=1}^J |\boldsymbol{x}^{(j)}(i) - \boldsymbol{x}^{(k)}(i)| \right), \tag{43}$$

which can be implemented in $O(nJ \log J)$ complexity using the equivalent form introduced in Zamo & Naveau (2018). In our experiments, we report the CRPS averaged over all test cases.

**Spread-skill ratio (SSR)**   The spread-skill ratio (SSR) is a simple yet powerful diagnostic of how well an ensemble prediction's stated uncertainty (spread) matches its actual error (skill) (Fortin et al., 2014). Intuitively, if the ensemble distribution truly captures the variability of the ground truth, then ensemble members should be statistically indistinguishable from observed outcomes. Formally, let $\{\boldsymbol{x}_i^*\}_{i=1}^N$ denote a set of observed ground truths. Suppose, for each observed ground truth $\boldsymbol{x}_i^*$, we have an ensemble prediction $\{\boldsymbol{x}_{i,j}\}_{j=1}^J$. Let $\bar{\boldsymbol{x}}_i = \frac{1}{J}\sum_j \boldsymbol{x}_{i,j}$. The unbiased estimator of SSR can be written as

$$\text{SSR} = \sqrt{\frac{\text{spread}^2}{\text{skill}^2}}, \tag{44}$$

where

$$\text{spread}^2 = \frac{1}{N}\sum_{i=1}^N \frac{1}{J-1}\sum_{j=1}^J \|\boldsymbol{x}_{i,j} - \bar{\boldsymbol{x}}_i\|_2^2,$$

$$\text{skill}^2 = \frac{1}{N}\sum_{i=1}^N \|\frac{1}{J}\sum_j \boldsymbol{x}_{i,j} - \boldsymbol{x}_i^*\|_2^2 + \frac{1}{J(J-1)}\text{spread}^2.$$

A value of $\text{SSR} = 1$ indicates the perfect calibration. Small SSR means that the ensemble prediction is over-confident while large SSR indicates that the ensemble prediction is over-cautious.

**Rank histogram**   The rank histogram (Anderson, 1996; Hamill, 1997; Talagrand, 1999) assesses ensemble calibration by comparing the truth to the empirical distribution formed by the ensemble. For each grid point in each test case, the ensemble members are sorted, and the rank of the true value within this ordering (0 means below all, $J$ means above all, or an intermediate integer) is recorded. Pooling these ranks over all points and cases yields a histogram with $J + 1$ bins. A flat histogram indicates a statistically consistent ensemble: the truth behaves like an additional random draw from the predictive distribution. A U-shape signals under-dispersion (ensemble spread too narrow), an inverted U indicates over-dispersion, and tilted shapes reveal bias.

### C.2.3   BASELINE IMPLEMENTATION

For methods that require training on paired data, specifically the end-to-end U-Net and conditional diffusion model (CDM), we first generate a collection of observation-solution pairs by simulating observations from the prior training dataset available in InverseBench (Zheng et al., 2025b). To evaluate their in-distribution performance, we retrain the U-Net and CDM for each noise level, which takes around 7-10 hours on a single GH200.

**U-Net**   The end-to-end U-Net baseline is adapted from the U-Net used in our diffusion model by removing the time conditioning branch. The observation is upsampled to the same resolution before being fed into the U-Net. However, it is important to note that observations are not always spatially aligned with the unknown signal in a general setting. Consequently, end-to-end neural networks typically require additional design considerations for different types of observations.

**CDM-CA**   CDM-CA is adapted from the U-Net architecture of the prior diffusion model. This involved replacing the self-attention module with cross-attention and incorporating a CNN-based observation encoder, following the conditioning mechanism used in Rombach et al. (2022).

**CDM-Cat**   CDM-Cat is adapted from the U-Net of the prior diffusion model. We upsample the observation to the same resolution as the solution and directly concatenate it with the input over the channel dimension. This conditioning mechanism only works for observations that are spatially aligned to the input.

To find the right hyperparameters for the training-free methods, we perform two rounds of Bayesian optimization over their hyperparameters on the validation set provided in InverseBench (Zheng et al., 2025b).

**Existing baselines in InverseBench**   We simply take the existing implementations from InverseBench (Zheng et al., 2025b) for baseline methods that already exist in the codebase including DPG (Tang et al., 2024), SCG (Huang et al., 2024), EnKG (Zheng et al., 2025a), and EKI (Iglesias et al., 2013).

**EKS + DM**   The original EKS (Garbuno-Inigo et al., 2020a) method only considers the Gaussian prior. The most straightforward way to incorporate the diffusion prior is to initialize the ensemble members with random samples from diffusion model. We fix the ensemble size to 1024 and optimize the other hyperparameters using Bayesian optimization.

**EKS w/ DP**   While the original update rules of EKS is derived for Gaussian prior only, we can approximate the gradient of the prior regularization term in EKS by evaluating the diffusion model $s_\theta(\boldsymbol{x}_t; \sigma(t))$ with $t$ close to zero. We fix the ensemble size to 1024 and optimize the other hyperparameters using Bayesian optimization.

**Localized EKS w/ DP**   Localization is a common technique to improve the multi-modal sampling of ensemble methods by assigning a localized mean and covariance for each particle. We implement the localized EKS w/ DP following the formulation in (Reich & Weissmann, 2021; Wagner et al., 2022). Again, we fix the ensemble size to 1024 and optimize the other hyperparameters using Bayesian optimization.

**FKD**   Feynman-Kac diffusion steering (FKD) (Zhao et al., 2025; Singhal et al., 2025) is a method based on interacting particle system, which can be viewed as a generalization of SCG (Huang et al.,

2024). We fix the ensemble size to 1024 and optimize the other hyperparameters using Bayesian optimization.

## C.3 IMAGE RESTORATION

### C.3.1 PROBLEM SETUP

We evaluated our method on image restoration tasks with the FFHQ256 dataset. Our evaluation set consisted of the first ten images, indexed 00000 to 00009, in the validation set. The pre-trained model is taken from Chung et al. (2023) (FFHQ256) and converted into an EDM checkpoint with their Variance-Preserving (VP) preconditioning (Karras et al., 2022). In general, we follow the experiment setup of Chung et al. (2023). All problem settings use a measurement noise of $\sigma_y = 0.05$. We address that PnP-DM results on phase retrieval are significantly worse than those originally reported. Note that we used a different forward model configuration, larger measurement noise, and full-color images, which differs from the original PnP-DM setup. Furthermore, PnP-DM super-resolution results are lower than originally reported. We note that we compare against the PnP-DM configuration that uses Langevin Monte Carlo during the likelihood step, which differs from their original configuration for linear inverse problems.

**Box Inpainting**    The forward model is given by

$$\boldsymbol{y} \sim \mathcal{N}(\boldsymbol{M} \odot \boldsymbol{x}, \sigma_y^2 \mathbf{I}),$$

where $\boldsymbol{M}$ is a binary masking matrix. We consider the case where the mask is a boxed region, which requires significantly stronger guidance from the prior for generation of plausible image content.

**Super-resolution**    The forward model is given by

$$\boldsymbol{y} \sim \mathcal{N}(\boldsymbol{P}_f \boldsymbol{x}, \sigma_y^2 \mathbf{I}),$$

where $\boldsymbol{P}_f$ is a linear operator that downsamples an image by a factor of $f$ with a block averaging filter. In our experiments, we set $f = 4$.

**Phase Retrieval**    The forward model is given by

$$\boldsymbol{y} \sim \mathcal{N}(|\boldsymbol{FP}\boldsymbol{x}|, \sigma_y^2 \mathbf{I}),$$

where $\boldsymbol{P}$ is an oversampling matrix and $\boldsymbol{F}$ is the Fourier transform. We set the oversampling ratio to 2.

### C.3.2 EVALUATION METRICS

**PSNR**    Peak Signal-to-Noise Ratio (PSNR) measures the ratio between the maximum power of a signal and the maximum power of the noise corrupting it. PSNR is a commonly used metric to assess the quality of image and video reconstruction. The PSNR between a prediction $\boldsymbol{x}$ and ground truth signal $\boldsymbol{x}^*$ is defined as

$$\text{PSNR} = 20 \cdot \log_{10}(\text{MAX}_{\boldsymbol{x}}) - 10 \cdot \log_{10}(\text{MSE}(\boldsymbol{x}, \boldsymbol{x}^*)), \tag{45}$$

where $\text{MAX}_{\boldsymbol{x}}$ is the maximum possible pixel value (i.e. 255).

**SSIM**    The structural similarity index measure (SSIM) (Wang et al., 2004) is another metric to compute the similarity between two images. It compares patterns of luminance, contrast, and structure between two images to achieve a metric more aligned with visual perception, given by

$$\text{SSIM} = \frac{(2\mu_{\boldsymbol{x}}\mu_{\boldsymbol{x}^*} + C_1)(2\sigma_{\boldsymbol{x}\boldsymbol{x}^*} + C_2)}{(\mu_{\boldsymbol{x}}^2 + \mu_{\boldsymbol{x}^*}^2 + C_1)(\sigma_{\boldsymbol{x}}^2 + \sigma_{\boldsymbol{x}^*}^2 + C_2)}, \tag{46}$$

where $\mu_{\boldsymbol{x}}, \mu_{\boldsymbol{x}^*}$ are the mean luminances, $\sigma_{\boldsymbol{x}}, \sigma_{\boldsymbol{x}^*}$ the the variances between images $\boldsymbol{x}$ and $\boldsymbol{x}^*$, respectively. The terms $C_1$ and $C_2$ are small constants to stabilize the computation and $\sigma_{\boldsymbol{x}\boldsymbol{x}^*}$ the covariance between the two images.

**LPIPS** The Learned Perceptual Image Patch Similarity (LPIPS) (Zhang et al., 2018) compares two images by measuring their distance in the feature space of a pre-trained neural network. LPIPS is computed as a weighted sum of the Euclidean distances between the network activations of an image $\boldsymbol{x}$ and an image $\boldsymbol{x}^*$.

## D ADDITIONAL EXPERIMENTS

**Linear Gaussian problems** We provide additional quantitative results on linear Gaussian problems. We report the SWD and $D_{\mathrm{KL}}$ for each method across different problem dimensions and observation noise levels in Table 4. As shown, Blade, EKS and MCGDiff (Cardoso et al., 2024) both achieve similar and strong performance, surpassing the other tested baselines. This aligns with theoretical expectations, given that these methods are proven to be asymptotically accurate in linear Gaussian problems. Consequently, the results empirically confirm that Blade can achieve accurate posterior sampling when applied to linear problems.

**Image restoration inverse problems** As complementary evidence of Blade's breadth, we evaluate Blade on three standard image restoration problems on the FFHQ256 dataset: inpainting and super-resolution (linear), and phase retrieval (nonlinear). While image restoration is not a primary goal of this work, we provide such experiments to establish a reference point against other common methods, as reported in Table 5. Experimental details can be found in Appendix C.3.1, and further qualitative results are in Appendix D. Notably, Blade shows robustness to the highly ill-posed, nonlinear phase retrieval forward model. We provide additional qualitative results on box inpainting, super-resolution, and phase retrieval in Figure 15.

**Black hole imaging** We also represent results of Blade on black hole imaging problem from InverseBench (Zheng et al., 2025b) in Table 3. As shown, Blade attains the best reconstruction quality (PSNR) among compared methods while remaining competitive on the EHT closure statistics. Additionally, we note that InverseBench assumes a single ground truth and does not evaluate posterior calibration or multi-modality. Since the goal in this work is calibrated posterior sampling, we view these additional results as complementary evidence.

Table 3: Evaluating Blade on the black hole imaging benchmark from InverseBench (Zheng et al., 2025b). Note that this benchmark only uses deterministic metrics with a focus on accurate point estimate. Blade's goal is well-calibrated posterior sampling. We therefore regard the results as complementary evidence.

| Method | PSNR | Blur PSNR | $\chi^2_{cp} \to 1$ | $\chi^2_{camp} \to 1$ |
|---|---|---|---|---|
| Blade(diag) | 30.83 | 36.22 | 2.12 | 1.61 |
| Blade(main) | 31.30 | 36.81 | 2.24 | 1.95 |
| DPS | 25.86 | 32.94 | 8.76 | 5.46 |
| LGD | 21.22 | 26.06 | 13.24 | 13.22 |
| PnPDM | 26.07 | 32.88 | 1.31 | 1.20 |
| DAPS | 25.60 | 32.78 | 1.30 | 1.23 |
| RED-Diff | 23.77 | 29.13 | 1.85 | 2.05 |
| DiffPIR | 25.01 | 31.86 | 3.27 | 2.97 |

## E ADDITIONAL DISCUSSION

Our current theoretical analysis are performed in the continuous-time and large particle limit although we provide the discretization and finite-particle dynamics for the practical implementation. Further investigation involving discretization error and specific discretization schemes is a valuable direction for future work. We expect the proposed Bayesian inversion algorithm to have a positive social impact in areas of science and engineering since it provides more reliable uncertainty calibration as shown

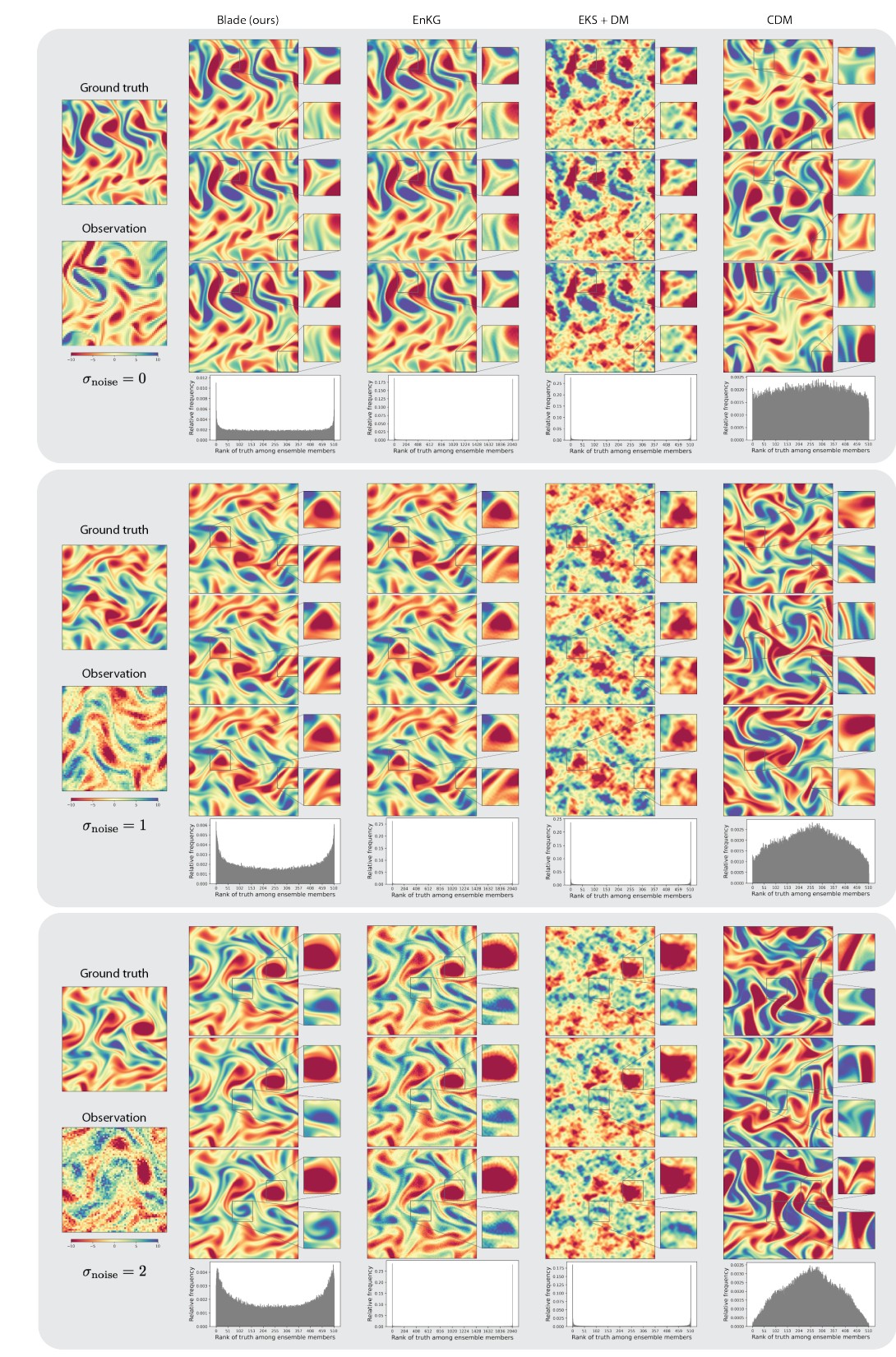

Figure 9: Qualitative and probabilistic assessment on Navier-Stokes inverse problem at three observation noise levels. Each block presents qualitative results with zoom-in views and the rank histogram calculated over all test cases. Blade produces the flow structures accurately while capturing realistic local variability, and its rank histogram demonstrates superior ensemble calibration.

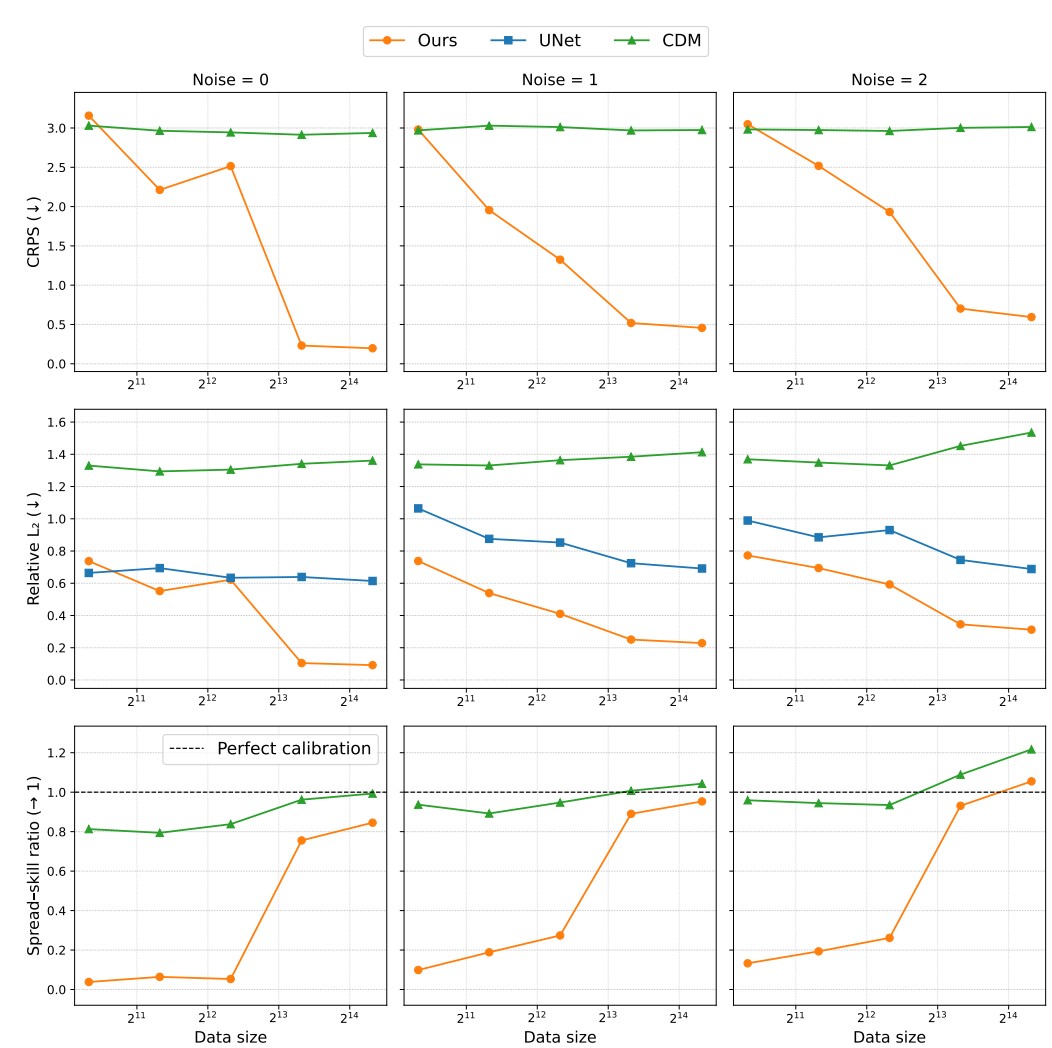

Figure 10: Scaling of different methods' performance with training data size across three measurement noise levels. Each row corresponds to one evaluation metric. U-Net produces deterministic prediction so the probabilistic metrics like CRPS and SSR do not apply. Note that our method uses the same diffusion prior across different noise levels while the training-based methods train separate networks for each noise level.

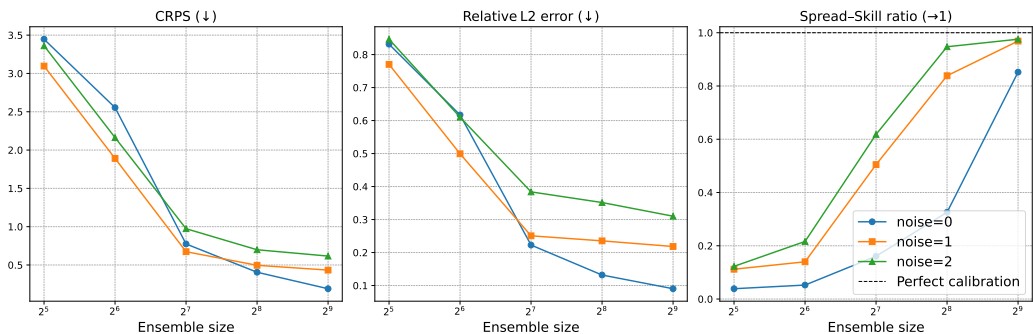

Figure 11: Effect of ensemble size. At each observation noise level, we study how the performance of `Blade` scales with the ensemble size.

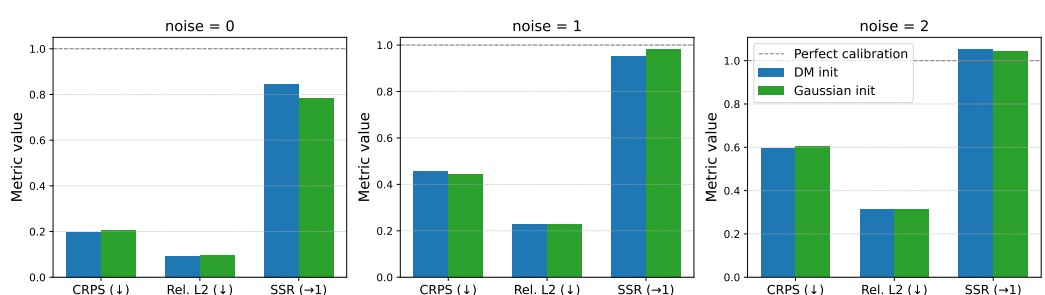

Figure 12: Effect of initialization. At each observation noise level, we compare the performance of `Blade` initialized from diffusion prior and Gaussian.

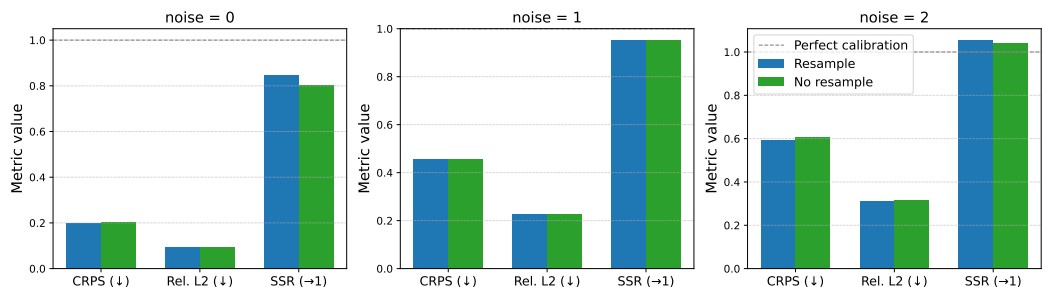

Figure 13: Effect of resampling strategy. At each observation noise level, we compare the performance of `Blade` with and without the resampling strategy.

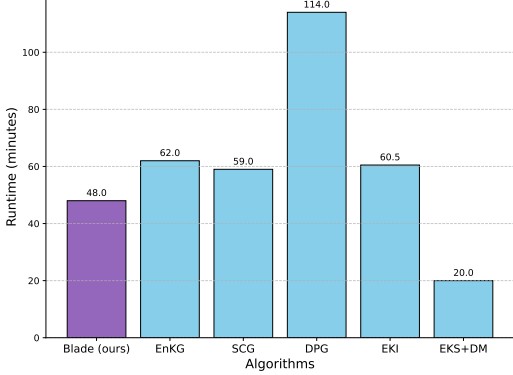

Figure 14: Runtime comparison of different algorithms on Navier-Stokes inverse problem, measured on a single GH200.

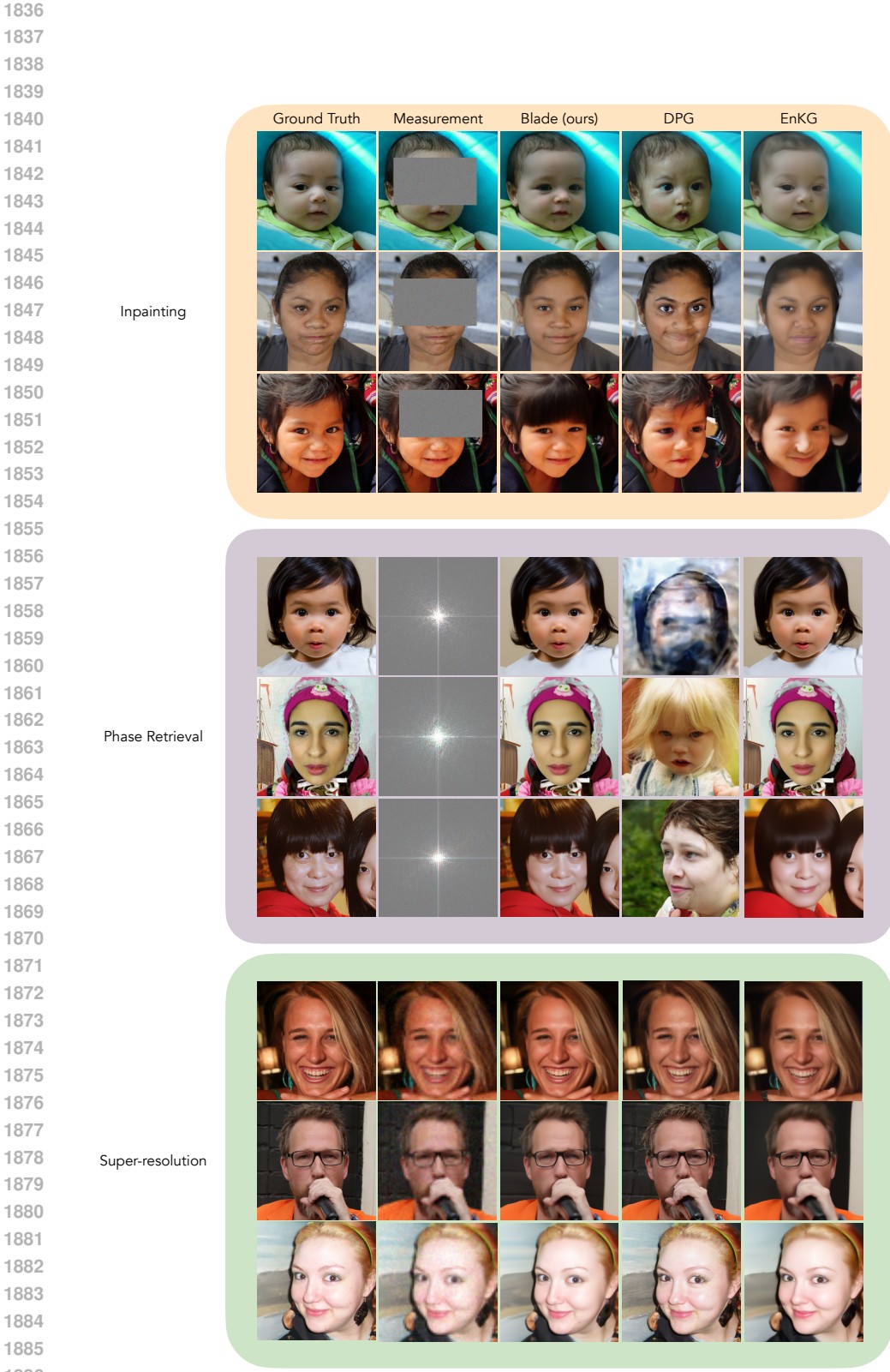

Figure 15: Additional image generation results.

Table 4: Experimental results on linear Gaussian problems for Sliced Wasserstein Distance (SWD) and KL Divergence ($D_{\mathrm{KL}}$) across different $\sigma_y$ values and methods for various data dimension $n$.

| $n$ | Method | $\sigma_y$ | | | | | | | |
|---|---|---|---|---|---|---|---|---|---|
| | | 0.5 | | 1.5 | | 2.5 | | 3.5 | |
| | | SWD | $D_{\mathrm{KL}}$ | SWD | $D_{\mathrm{KL}}$ | SWD | $D_{\mathrm{KL}}$ | SWD | $D_{\mathrm{KL}}$ |
| 2 | DPG | 4.199 | 11.83 | 3.955 | 122.11 | 3.969 | 350.76 | 4.646 | 1219.96 |
| | SCG | 2.826 | 85k | 2.704 | $\geq$ 100k | 3.072 | $\geq$ 100k | 3.814 | $\geq$ 100k |
| | EnKG | 1.832 | $\geq$ 100k | 1.752 | $\geq$ 100k | 1.972 | $\geq$ 100k | 3.020 | $\geq$ 100k |
| | EKS | 1.651 | **0.374** | 2.072 | 0.493 | 2.061 | 0.502 | 2.204 | 0.549 |
| | MCGdiff | **1.423** | 1.002 | **1.497** | 1.238 | **1.511** | 0.899 | 1.760 | 0.985 |
| | Blade (Ours) | 1.915 | 0.556 | 1.725 | **0.348** | 1.763 | **0.423** | **1.678** | **0.381** |
| 80 | DPG | 6.786 | 69.66 | 7.005 | 67.32 | 6.905 | 66.11 | 7.289 | 67.13 |
| | SCG | 6.022 | 1708. | 6.059 | 15916. | 6.033 | 37121. | 6.013 | 91466. |
| | EnKG | 4.997 | $\geq$ 100k | 4.938 | $\geq$ 100k | 5.180 | $\geq$ 100k | 5.068 | $\geq$ 100k |
| | EKS | **2.444** | **27.72** | **2.357** | **25.91** | **2.366** | **25.69** | **2.371** | **25.60** |
| | MCGdiff | 33.03 | 177.3 | 32.90 | 177.2 | 32.87 | 177.2 | 32.93 | 177.5 |
| | Blade (Ours) | 4.367 | 64.13 | 4.492 | 65.99 | 4.284 | 67.65 | 4.579 | 61.06 |
| 400 | DPG | 6.111 | $\geq$ 100k | 6.149 | $\geq$ 100k | 6.259 | $\geq$ 100k | 6.181 | $\geq$ 100k |
| | SCG | 6.199 | $\geq$ 100k | 6.172 | $\geq$ 100k | 6.182 | $\geq$ 100k | 6.276 | $\geq$ 100k |
| | EnKG | 8.636 | $\geq$ 100k | 7.432 | $\geq$ 100k | 11.513 | $\geq$ 100k | 11.825 | $\geq$ 100k |
| | EKS | **1.047** | **1817.** | **1.092** | 1777. | **1.114** | 2156. | **1.130** | 2134. |
| | MCGdiff | 33.09 | $\geq$ 100k | 33.08 | $\geq$ 100k | 32.98 | $\geq$ 100k | 33.06 | $\geq$ 100k |
| | Blade (Ours) | 4.414 | 2478. | 4.527 | **1747.** | 4.074 | **1794.** | 4.340 | **1805.** |

Table 5: Qualitative evaluation on FFHQ 256x256 dataset. We report average metrics for image quality and samples consistency on three tasks. Measurement noise level $\sigma = 0.05$ is used if not otherwise stated. (†: the general PnP-DM algorithm that uses Langevin dynamics for likelihood step.)

| | Inpaint (box) | | | SR ($\times 4$) | | | Phase retrieval | | |
|---|---|---|---|---|---|---|---|---|---|
| | PSNR↑ | SSIM↑ | LPIPS↓ | PSNR↑ | SSIM↑ | LPIPS↓ | PSNR↑ | SSIM↑ | LPIPS↓ |
| **Gradient access** | | | | | | | | | |
| DPS | 21.77 | 0.767 | 0.213 | 24.90 | 0.710 | 0.265 | 16.79 | 0.589 | 0.448 |
| PnP-DM† | 22.17 | **0.832** | **0.136** | 25.86 | 0.808 | 0.193 | 18.98 | 0.650 | 0.409 |
| **Black-box access** | | | | | | | | | |
| DPG | 20.89 | 0.752 | 0.184 | 28.12 | **0.831** | **0.126** | 8.76 | 0.297 | 0.663 |
| EnKG | 21.70 | 0.727 | 0.286 | 27.17 | 0.773 | 0.237 | 24.02 | 0.796 | 0.232 |
| Blade (Ours) | **23.70** | 0.763 | 0.225 | **29.01** | 0.826 | 0.204 | **25.99** | **0.839** | 0.215 |

in our experiments, particularly for problems like weather data assimilation where the uncertainty calibration is crucial.

