# OpenReview forum: "Blade: A Derivative-free Bayesian Inversion Method using Diffusion Prior"
_ICLR.cc/2026/Conference — Submitted to ICLR 2026_

### Official Review · Reviewer_jPCJ · 2025-10-24

**Soundness:** 2
**Presentation:** 3
**Contribution:** 2
**Rating:** 4
**Confidence:** 4

**Summary:**

This paper introduces Blade, a derivative-free Bayesian inversion method that uses a pretrained diffusion model as a data-driven prior. It builds on the split Gibbs sampler, alternating between (i) a derivative-free likelihood step realized via an ensemble-based statistical linearization of the black-box forward model, and (ii) a prior step implemented as the reverse denoising diffusion process. The authors provide non-asymptotic convergence/error bounds that separate the contributions of the forward-model linearization and the diffusion-prior score approximation. Empirically, Blade is evaluated on linear–Gaussian (and Gaussian-mixture) settings and on a challenging Navier–Stokes fluid-dynamics inverse problem

**Strengths:**

Diffusion models continue to gain popularity, yet conditional sampling with diffusion priors remains a challenging and important problem, especially when they are used as prior distributions in Bayesian inference. This paper addresses that gap by proposing an ensemble-based particle sampling approach that effectively combines two active research directions: particle-based Bayesian inference and diffusion-model priors.

While prior work on diffusion-based split Gibbs sampling has primarily focused on improving the prior step, the authors make a novel contribution by developing a more efficient and theoretically grounded likelihood step. They further strengthen their proposal by quantifying the effects of key approximation errors, those arising from statistical linearization of the forward model and score approximation in the diffusion prior.

Finally, the paper offers an extensive empirical evaluation across both controlled and complex nonlinear inverse problems, providing convincing evidence for the effectiveness and robustness of the proposed algorithm.

**Weaknesses:**

- My main concern lies with the error bound in Theorem 2. The authors bound the discrepancy between systems (9) and (6), but both are assumed to share the same sample covariance $C_t$ computed from the particles of system (9). This assumption greatly simplifies the analysis and, strictly speaking, does not capture the true approximation error between the two dynamics. Instead, it effectively introduces an auxiliary process (6.5) that depends on $C_t$ from (9). The setup is further confusing because system (6) is initialized from its stationary distribution, while (9) is not. A more rigorous comparison would require analyzing how the respective covariance matrices evolve over time.

- The convergence proofs offer only limited novelty and appear to be incremental extensions of known results. Lemma 1 reproduces existing results (as acknowledged), Theorem 1 follows standard arguments under the assumption that $C_t$ remains strictly positive definite, and Lemma 2 closely mirrors Lemma A.4 from Wu et al. (2024). Similarly, the proof structure of Theorem 2 largely parallels Theorem 3.1 in Wu et al. (2024). Lemma 3 is again adapted from previous work. Overall, while the analysis is sound, its theoretical originality is limited.

- The assumptions underlying Theorem 2 are not adequately discussed in the main text, and the appendix treatment is brief. In particular, Assumption 3 is not mentioned outside the appendix, and Assumption 2 seems quite restrictive. Typically, the linearization error can be bounded at a fixed time via a Taylor expansion, yielding an error term dependent on the ensemble covariance norm. However, it is unclear whether such a bound remains uniformly valid over time. A more detailed justification, or an explicit derivation for the proposed algorithm, would strengthen the theoretical section considerably.

- The paper does not include a comparison to the EKS or ALDI variants that incorporate localization (see, e.g., Fokker-Planck Particle Systems for Bayesian Inference: Computational Approaches, Reich & Weissmann, 2021). Localization has been shown to substantially improve the performance of EKS/ALDI in multimodal settings, and it can be implemented efficiently using particle clustering (see, e.g., The Ensemble Kalman Filter for Rare Event Estimation, Wagner et al., 2022). Including such comparisons would be important, especially since EKS appears to outperform Blade in the linear–Gaussian case (Table 4) while also having a lower runtime.

**Questions:**

Below I list my questions and suggestions for the authors:

- How does Blade perform compared to advanced variants of ALDI or EKS that use localized covariance preconditioners?
- How exactly are ALDI and EKS implemented when incorporating diffusion priors? Are you using the estimated score from the diffusion model as an approximation of the gradient of $g$?
- In the evaluation, the paper reports only statistics from the final iteration. How does performance evolve over the number of iterations? How do you assess convergence?
- I recommend discussing Assumptions 1-3 directly in the main text rather than deferring them entirely to the appendix.
- The discussion of theoretical novelty (e.g., in Remark 2) could be expanded. While similarities to prior work are acknowledged, the current phrasing is somewhat vague. Please clarify the specific conceptual or technical differences relative to earlier analyses (e.g., Wu et al., 2024).
- Line 48: Derivative-free or zeroth-order gradient approximations also tend to scale poorly in high dimensions. Do you have theoretical or empirical evidence suggesting that statistical linearization scales more favorably?
- Line 321: Doesn’t the convergence bound require the assumption that both $\lambda^\ast$ and $\delta$ are strictly positive?
- Figure 11: It is somewhat surprising that increasing the ensemble size beyond a certain point does not improve performance. Are other error sources dominating at that stage? Would the observed plateau decrease if you retuned hyperparameters (e.g., step size, noise level) for larger ensembles?

Typos and minor comments:

- Line 138: we follow[]
- Line 209: distributio[n]
- Line 231: to run [the] algorithm
- Line 333: strictly speaking the statistical linearization error would be avoidable by computing derivatives (if available).
- Line 332: Theorem 1 show[s]
- Line 345: This statement is a bit vague, whether the posterior distribution yields a good solution for the ill-posed inverse problem depends on multiple choices. I think it is clear, that you evaluate your algorithm Blade on statistics of the posterior samples since you are working in a Bayesian setup.
- Line 475: descent should be decent
- Section 4.2: U-Net vs UNet
- Line 1013: „… on on …“
- Line 1020: „…, [a]s shown in…“
- Line 1107: Missing space „…performance. []We denote…“

---

> ### Author Response · Authors · 2025-12-01
> **Response to "error bound in Theorem 2"**
>
> We thank the reviewer for the thoughtful comment. We agree that it is natural to attempt to analyze the distance between two processes with their own covariance evolutions. However, our analysis intentionally isolates a different aspect of the problem.
> - As established in Theorem 1, the idealized process  gives the target distribution we want in Eq.(3). In particular, this process starting from stationary distribution would stay at the stationary process if the likelihood step is exact regardless of the chosen $C_t$. The role of preconditioner $C_t$ is to specify the geometry with respect to which we measure approximation error (See Kalman-Wasserstein gradient flow in [1] for how geometry depends on the preconditioner). Therefore, fixing the $C_t$ across both processes does not alter the target stationary distribution; it simply allows us to compare the two dynamics within a common metric structure. Hence, **the convergence to the reference process indicates the convergence to the desired stationary distribution.**
> - Theorem 2 aims to show 1) the convergence to this reference process under the same geometry 2) stability under statistical linearization error and score approximation error.
> - Comparing two systems with their own evolving preconditioners introduces a moving metric on each side. Bounding the distance betwen two processes under different and time-varying metrics is not only technically difficult, but more importantly, it is unclear what notion of distance under such geometric mismatch would be meaningful.
>
> We have added a paragraph (line 365-370) in the updated manuscript to clarify this.
>
> [1]: Garbuno-Inigo, Alfredo, et al. "Interacting Langevin diffusions: Gradient structure and ensemble Kalman sampler." SIAM Journal on Applied Dynamical Systems 19.1 (2020): 412-441.

---

> ### Author Response · Authors · 2025-12-01
> **Response to "convergence proof offers limited novelty"**
>
> We thank the reviewer for the feedback. We agree that our analysis builds on several existing proof techniques, but our analysis **strictly** generalizes the prior result to a more realistic and technically challenging regime. It contains the following conceptual differences compared to prior work such as Wu et al. (2024):
> - **Different and more complex dynamics**: we consider an interacting particle system with state-dependent preconditioner within split Gibbs framework whereas Wu et al analyzes the standard Langevin dynamics. This requires non-trivial technical extensions:
>     - Generalizing of Lemma A.4 of Wu et al. to matrix-valued diffusion terms, which introduces additional dependencies on covariance eigenvalues that do not appear in Wu et al.
>     - Handle drift terms that lie in the range of ensemble covariance.
> - **More realistic and practically relevant setup** Wu et al. assume that the likelihood step is exact. Our setting is more realistic by considering finite time execution of the dynamics as well as the forward model error.
> - **Richer results**: Theorem 2 explicitly shows how the algorithm remains stable under forward model error and how the finite time horizons affect convergence, which are not addressed in Wu et al. 2024.

---

> ### Author Response · Authors · 2025-12-01
> **Response to "assumptions for Theorem 2"**
>
> We thank the reviewer for the feedback and suggestion. Assumptions and their discussions are defered to appendix due to page limit. For the rebuttal version (one extra page), we have moved the assumptions into the main text and expanded the accompanying discussion to improve clarity.
>
> Assumption 2 requires the statistical linearization error remains uniformly bounded over the time horizon of interest. As the reviewer noted, this can be bounded at a fixed time by the ensemble covariance (with standard regularity assumptions on forward model $\mathcal{G}$). A uniformly bounded ensemble covariance is realistic because in most practical scenarios we study, the ensemble covariance remains bounded over iterations.

---

> ### Author Response · Authors · 2025-12-01
> **Response to "how is EKS implemented when using diffusion prior" and "comparison to localized EKS"**
>
> > How exactly are ALDI and EKS implemented when incorporating diffusion priors?
>
> We assumes the baseline that reviewer is referring to is "EKS+DM" in Table 1. This is simply initializing the EKS particles by sampling from the diffusion model. We adopted this approach because the original EKS formulation assumes a Gaussian prior, and initialization from diffusion model samples seemed the most straightforward way to incorporate a learned prior.
>
> However, as the reviewer correctly points out, we can also use the diffusion model to approximate the score of the prior in the EKS dynamics. We agree this is a good baseline and have added it as "EKS w/ DP" (EKS with diffusion prior). We present this new baseline along with its localized version in the table below. The results in Fig. 1 and Table 1 are also updated accordinly. We found that while EKS w/ DP shows improvement over EKS + DM, it still underperforms the best existing baselines and exhibits poor calibration.
>
> > The paper does not include a comparison to the EKS or ALDI variants that incorporate localization
>
> We thank the reviewer for the reference and suggestion. We have added the baselines EKS w/ DP (using diffusion model as prior score) and localized EKS w/ DP to our comparison. The additional comparisons are shown below, and Figure 1 and Table 1 have been updated accordingly.
>
>
> | **noise** | **Method**          | **CRPS↓** | **SSR→1** | **Rel L2 error↓** |
> | --------- | ------------------- | --------- | --------- | ----------------- |
> | 0.0       | EKS+DM              | 1.900     | 0.181     | 0.539             |
> |           | EKS w/ DP           | 1.280     | 0.061     | 0.336             |
> |           | Localised EKS w/ DP | 1.643     | 0.056     | 0.428             |
> |           | EnKG                | 0.395     | 0.164     | 0.120             |
> |           | **Blade (main)**    | **0.216** | **0.955** | **0.110**         |
> | 1.0       | EKS+DM              | 2.088     | 0.218     | 0.606             |
> |           | EKS w/ DP           | 1.723     | 0.085     | 0.455             |
> |           | Localized EKS w/ DP | 1.887     | 0.081     | 0.495             |
> |           | EnKG                | 0.651     | 0.154     | 0.191             |
> |           | **Blade (main)**    | **0.453** | **0.950** | **0.229**         |
> | 2.0       | EKS+DM              | 2.255     | 0.280     | 0.685             |
> |           | EKS w/DP            | 2.094     | 0.080     | 0.547             |
> |           | Localized EKS w/ DP | 2.057     |  0.089    | 0.542             |
> |           | EnKG                | 1.032     | 0.144     | 0.294             |
> |           | **Blade (main)**    | **0.608** | **0.949** | **0.306**         |
>
> - Both EKS w/ DP and localized EKS w/ DP demonstrated imporved CRPS and relative L2 over EKS + DM
> - They are outperformed by the best existing baseline (EnKG) in Table 1
> - Blade consistently outperforms these baselines across different metrics and measurement noises.
>
> The reviewer correctly notes that EKS outperforms Blade in the linear-Gaussian setting. This is expected—EKS is specifically designed for Gaussian distributions and should excel in this regime. Blade's advantage emerges in the nonlinear, multimodal setting  (Navier-Stokes experiments) with complex data prior, where maintaining calibration and precision becomes challenging.
>
> Implementation details are added to Appendix C.2.3: we used 1024 particles and all hyperparameters were optimized via two rounds of Bayesian optimization on the InverseBench validation set.

---

> ### Author Response · Authors · 2025-12-01
> **Response to "How do you assess convergence?"**
>
> We assess convergence through ablation study on the number of iterations. As shown in Fig. 4, we sweep over the number of iterations and found that the performance (measured by CRPS, SSR, and relative L2 error) plateaus around 25 iterations. Increasing iterations beyond this point yields marginal gains, indicating convergence. Based on this analysis, we use 25 iterations throughout all experiments.We primarily track the probabilistic metrics CRPS and SSR to assess whether the ensemble has converged to a well-calibrated posterior approximation.

---

> ### Author Response · Authors · 2025-12-01
> **Response to "Zero-order gradient approximation vs statistical linearization"**
>
> We thank the reviewer for this important question. Statistical linearization differs fundamentally from standard zeroth-order methods in both purpose and scaling:
> - **Standard zeroth-order methods** (e.g., finite differences) are local approximations that estimate the gradient at each particle independently. This requires $O(n)$ per particle ($n$ is the dimensionality of the particle), resulting in $O(Jn)$ total complexity.
> - **Statistical linearization** is a global approximation that statistically linearizes the forward model over the ensemble with $O(J)$ complexity (hence the "interacting" particle system). This is not a substitute for gradient approximation at individual points; rather, it captures the collective behavior of the forward model over the ensemble distribution.
>
> **Advantage of statistical linearization** Locally accurate gradients (from finite differences or exact computation) can trap particles in local structures of the potential landscape (Figure 2, left panel). Statistical linearization smooths out these local features, allowing the ensemble to focus on the big structures—particularly valuable for posterior landscapes with many local traps.
>
> For example, if we look at the Fig. 2 in our paper, the local gradient may not be reliable in this case because the local structure is very jagged. In contrast, statistical linearization provides a smooth global structure without getting trapped in small local modes. Zheng et al. 2024 [1] experimented with the finite-difference idea (Central-GSG in Fig.2) and it does not work well.
>
> [1]: Zheng, Hongkai, et al. "Ensemble Kalman Diffusion Guidance: A Derivative-free Method for Inverse Problems." Transactions on Machine Learning Research.

---

> ### Author Response · Authors · 2025-12-01
> **Response to "increasing the ensemble size beyond a certain point does not improve performance"**
>
> We appreciate the reviewer's question. The decreasing rate of improvement in metrics is expected because the algorithm already produces high-quality posterior samples (Fig. 9) with 256-512 particles, so further improvement becomes progressively harder. For example, improving CRPS from 0.5 to 0.2 is much harder than reducing it from 2.0 to 1.0.
>
> The main sources of error that do not vanish with larger ensembles are
> - Statistical linearization error and score approximation error as shown in Theorem 2
> - Discretization error of the continuous dynamics of the algorithm
> - Discretization error of the PDE solver itself (forward model).
>
> However, we note that for the ablation in Fig. 11, all hyperparameters are intentionally kept fixed across ensemble sizes. Retuning for each ensemble size would conflate the effect of ensemble size with hyperparameter changes.

---

> ### Author Response · Authors · 2025-12-01
> **Response to "typos and minor comments"**
>
> We really appreciate your careful reading and comments. We have fixed all the typos in the updated manuscript. We have also updated Line 345 (first sentence of Sec. 4) as follows: "Inverse problems are often ill-posed, so an ensemble of posterior samples with calibrated uncertainty is often more desirable than a single point estimate."

---

> ### Author Response · Authors · 2025-12-01
> **Response to "Does theorem 2 require strictly positive $\lambda^\ast$ and $\delta$ "**
>
> Yes, Theorem 2 requires both $\lambda^\*$ and $\delta$ are strictly positive. We have made this explicit in Theorem 2 in the updated manuscript. These conditions are satisfied in practice: $\delta$ can be controlled to strictly positive through $\beta(t)$ in the Eq.(2) or the numerical truncation in time (standard practice in diffusion model). $\lambda^*$ is strictly positive as long as the ensemble does not collapse.

---

### Official Review · Reviewer_wxbH · 2025-10-31

**Soundness:** 3
**Presentation:** 4
**Contribution:** 3
**Rating:** 6
**Confidence:** 2

**Summary:**

The paper proposes Blade, a method for Bayesian inversion based on Diffusion Split Gibbs Sampling and statistical linearization.
Blade comes in two variants: main and diag with different estimations of the covariance. The authors give a convergence analysis with explicit error bounds.
Blade is evaluated on several benchmark tasks; in the main paper on Gaussian/Gaussian mixture models, and a Navier-Stokes problem;
and compared to both optimization-based and other ensemble methods.

**Strengths:**

- The paper is very well written and easy to follow.
- The statistical linearization step is well motivated and a sample-efficient solution for non-differentiable forward models.
- Explicit error bound that includes both errors from the diffusion and forward model derivative approximation
- Both variants diag and main are well motivated and target different scenarios (point-estimation vs. callibrated posterior). This is shown convincingly in the Navier Stokes experiment
- Comprehensive set of experiments and ablations in the appendix

**Weaknesses:**

- Incremental improvement to the diffusion-based split Gibbs sampling
- No comparisons to sequential MCMC approaches; e.g., Feynman-Kac diffusion steering [1,2] which are also based on interacting particles. This would strengthen the results.
- I am not really convinced the CDM model is implemented in the most optimal way; just concatenating the observation as an additional channel is simpler and likely to give improved results.
- One downside of the likelihood step is that the forward model is evaluated on noisy samples. If the forward model is robust to noise this still works, but many applications have highly non-linear forward models where the statistical linearization might not work very well.

Minor typos:
- L1020 ", As shown"
- L209 "distributio"


[1] https://arxiv.org/pdf/2409.09650v1

[2] https://arxiv.org/pdf/2501.06848

**Questions:**

- If the forward model G was differentiable, can we use the gradient information directly in eq (6)? How does the statistical linearization compare in this case? Are inference times comparable?

---

> ### Author Response · Authors · 2025-12-01
> **Response to "incremental improvement"**
>
> We respectfully disagree with the characterization of our work as incremental. Blade represents substantial algorithmic and theoretical contributions:
> - Blade is the first diffusion-based derivative-free inference algorithm that produces calibrated samples with theoreical guarantees that explicitly control errors from forward model and diffusion model. No previous diffusion-based methods offer this to our knowledge.
> - Algorithmically, Blade substantially departs from the vanilla split Gibbs sampling (SGS) methods. In particular, SGS methods run independent chains for each sample, but Blade iterates an interacting particle system that enables derivative-free inference. This allows Blade to handle problems where forward model gradients are unavailable or unreliable — a setting that previous SGS methods cannot address. Blade produces well-calibrated posterior samples with SOTA probabilistic scores as measured in Table 1.
> - Improved theoretical analysis: our theoretical analysis strictly generalizes the theory in prior work [3] to a more realistic and technically challenging regime.
>     - We consider an interacting particle system with state-dependent preconditioner within the split Gibbs framework, which is more complex than the standard Langevin dynamics in the piror work. This requires non-trivial technical extensions.
>     - Our analysis setup is more realistic and practically relevant by considering finite time execution of the dynamics as well as the forward model error. Theorem 2 explicitly shows how the algorithm remains stable under forward model error and how the finite time horizons affect convergence, which are not addressed in the prior work.
>
> [3]: Wu, Zihui, et al. "Principled probabilistic imaging using diffusion models as plug-and-play priors." Advances in Neural Information Processing Systems 37 (2024): 118389-118427.

---

> ### Author Response · Authors · 2025-12-01
> **Response to "comparison to Feyman-Kac diffusion steering"**
>
> Thank you for these valuable references. We agree that Feynman-Kac diffusion steering [1,2] provides an important comparison as another interacting particle method. We have added this baseline (FKD-Steering) to Table 1.
> Our results show that FKD-Steering struggles to produce calibrated ensemble even with extensive hyperparameter search and is outperformed by some existing baselines included in Table 1 and significantly worse than Blade.
>
> | **noise** | **Method**       | **CRPS↓** | **SSR→1** | **Rel L2 error↓** |
> |-------------|-------------------|-----------|-----------|--------------------|
> | **0**       | FKD-Steering      | 1.604     | 0.002     | 0.399              |
> |             | EKS w/ DP     | 1.280        | 0.061        | 0.336              |
> |             | EnKG               | 0.395         | 0.164         | 0.120              |
> |             | **Blade (main)**  | **0.216** | **0.955** | **0.110**          |
> | **1**       | FKD-Steering      | 1.416     | 0.050     | 0.368              |
> |             | EKS w/DP               | 1.723        | 0.085        | 0.455             |
> |             | EnKG               | 0.651         | 0.154         | 0.191              |
> |             | **Blade (main)**  | **0.453** | **0.950** | **0.229**          |
> | **2**       | FKD-Steering      | 1.810     | 0.012     | 0.455              |
> |             | EKS w/DP               | 2.094         | 0.080        | 0.547              |
> |             | EnKG               | 1.032         | 0.144         | 0.294              |
> |             | **Blade (main)**  | **0.608** | **0.949** | **0.306**          |
>
>
> Implementation details: We used 1024 particles for FKD-Steering. All hyperparameters were optimized via two rounds of Bayesian optimization on the InverseBench validation set. Details are provided in Appendix C.2.3.

---

> ### Author Response · Authors · 2025-12-01
> **Response to "CDM with channel concatenation"**
>
> Thank you for this suggestion. In our initial submission, CDM was conditioned through cross-attention to support general measurement operators beyond spatially-aligned observations. However, we agree that channel concatenation is a simpler and more natural baseline for the Navier-Stokes problem where measurements are spatially aligned.
>
> We have added this baseline (CDM-Cat) to Table 1. As the reviewer anticipated, channel concatenation does improve performance over the cross-attention version (CDM-CA) but still significantly worse than Blade. The updated results are shown below:
>
>
> | noise | **Method**       | **CRPS ↓** | **SSR → 1** | **Rel L2 error ↓** |
> | ----- | ---------------- | ---------- | ----------- | ------------------ |
> | 0     | CDM-CA           | 2.900      | 0.983       | 1.362              |
> |       | CDM-Cat          | 1.413      | 0.896       | 0.653              |
> |       | **Blade (main)** | **0.216**      | **0.955**       | **0.110**              |
> | 1     | CDM-CA           | 2.872      | 1.059       | 1.409              |
> |       | CDM-Cat          | 1.805      | 0.979       | 0.873              |
> |       | **Blade (main)** | **0.453**      | **0.950**       | **0.229**              |
> |  2     | CDM-CA           | 2.993      | 1.087       | 1.542              |
> |       | CDM-Cat          | 2.211      | 0.974       | 1.043              |
> |       | **Blade (main)** | **0.608**      | **0.949**       | **0.306**              |

---

> ### Author Response · Authors · 2025-12-01
> **Response to "noisy sample"**
>
> The forward model in our main experiment (Navier-Stokes) is precisely the type of challenging case the reviewer mentions—a PDE solver that is highly nonlinear and requires stability conditions on inputs to function properly. Despite this, Blade works reliably without stability issues across all our experiments. Note that Blade provides explicit control over the noise level in the likelihood step through the coupling factor $\rho_k$.

---

### Official Review · Reviewer_vs1A · 2025-11-01

**Soundness:** 4
**Presentation:** 4
**Contribution:** 3
**Rating:** 8
**Confidence:** 4

**Summary:**

The authors established a non-asymptotic convergence analysis to characterize the impact of forward modeling and prior estimation errors. Experimental results show that Blade outperforms existing derivative-free Bayesian inversion methods on various inverse problems, including highly challenging highly nonlinear hydrodynamic problems.

**Strengths:**

Both the theoretical and experimental parts are excellent.

**Weaknesses:**

I think the experiment could be more thorough.

**Questions:**

Are you considering experiments on a larger scale?

---

> ### Author Response · Authors · 2025-12-01
>
> Thank you for your positive feedback and question.
>
> The scale and dimension of our experiments match and even exceed prior work like [1]:
> the Navier-Stokes experiments operate at dimension 16384 following InverseBench [2] (Figure 1, Table 1, Figure 9) and extended experiments on image restoration reach dimension 197k (Table 5, Figure 15). This scale provides sufficient complexity to meaningfully differentiate algorithm performance while remaining computationally tractable for the extensive ablations and comparisons needed to thoroughly validate a new method.
>
> We are also engaging with domain collaborators to work towards operational weather problem [3], which involves significantly larger state dimensions and real-world data complexities. However, such project requires non-trivial engineering and validation effort is beyond the scope of this paper, which focuses on introducing and analyzing a new algorithm.
>
> [1]: Wu, Zihui, et al. "Principled probabilistic imaging using diffusion models as plug-and-play priors." Advances in Neural Information Processing Systems 37 (2024): 118389-118427.
>
> [2]: Zheng, Hongkai, et al. "InverseBench: Benchmarking Plug-and-Play Diffusion Priors for Inverse Problems in Physical Sciences." The Thirteenth International Conference on Learning Representations.
>
> [3]: Morcrette, Jean-Jacques, CH JAKOB, and J. TEIXEIRA. "European Centre for Medium range Weather Forecasts." Numerical Modeling of the Global Atmosphere in the Climate System 550 (2000): 263.

---

### Official Review · Reviewer_NR9p · 2025-11-01

**Soundness:** 2
**Presentation:** 2
**Contribution:** 1
**Rating:** 2
**Confidence:** 5

**Summary:**

- The authors present a method for posterior sampling that leverages denoising diffusion models as priors.
- The proposed framework addresses settings where only pointwise evaluations of the likelihood are available.
- For that, the authors introduce a Split-Gibbs sampling scheme that alternates between sampling from the diffusion prior and a likelihood update.
- To circumvent the need for likelihood gradients, they employ a covariance-preconditioned Langevin Dynamics approach, in which the drift term is approximated via statistical linearization using an ensemble of particles.
- The proposed method is validated on a range of inverse problems, including both synthetic and real setup

**Strengths:**

- Introducing a method that combines statistical linearization and Split-Gibbs sampling to solve inverse problems with diffusion priors without requiring a gradient of the likelihood

**Weaknesses:**

**Motivation of the method**
The introduction of the covariance-preconditioned Langevin Dynamics appears somewhat arbitrary and insufficiently justified. Its use seems primarily motivated by convenience to avoid directly inverting the covariance matrix $C_t$, rather than by a clear theoretical or empirical rationale.

On the other hand, equation (9) involves the square root of $C_t$, which as such is difficulty to handle in practice, and the proposed method to handle it is ambiguous (see the point below on Correctness).


**Evaluation of the method**
The experimental validation is limited and does not support the method’s intended use case.
The approach is motivated by scenarios where the likelihood gradient is unavailable or costly to compute, yet all considered experiments (toy problems, Navier–Stokes, and black hole imaging) involve forward models for which gradients can be readily obtained; see [1] for Navier–Stokes and [2] for black hole imaging.
Consequently, the evaluation does not demonstrate the method’s effectiveness in the truly derivative-free regime.
Furthermore, the reported runtime in Table 14 is very computationally heavy (around 1 hour per reconstruction )and as such the method does not offer clear performance benefits over gradient-based alternatives.

**Correctness**
The use of the square-root covariance matrix approximation in Line 269 is incorrect, the approximation has shape $n \times$ the number of particles $J$, but is intended to condition the noise $dw_t \in \mathbb{R}^n$. In particular, for the single-particle case, $\sqrt{C_t}$ becomes a column matrix, making the matrix–vector product undefined.

**Typos and minor issues**
* Equation (2) introduces a nonstandard $\beta(t)$ coefficient that does not align with conventional diffusion model formulations, and I cannot find it in the cited reference [5]
* Line 86: replace "sampling for posterior inference" ---> “or sampling for inference.”
* Line 209: "distributio" ---> "distribution"
* In Theorem 1 and 2 the term "horizon" is ambiguous

**Missing references and related work**
- The statement in Lines 90–91 overlooks prior sampling-based approaches that handle nonlinear setups, such as [3].
- The discussion on Gibbs sampling and diffusion priors should cite [4], which provides a relevant treatment of Gibbs-sampling and inverse problems with diffusion models.


---

... [1] Rozet, François, and Gilles Louppe. "Score-based data assimilation." Advances in Neural Information Processing Systems 36 (2023): 40521-40541.

... [2] Wu, Zihui, et al. "Principled probabilistic imaging using diffusion models as plug-and-play priors." Advances in Neural Information Processing Systems 37 (2024): 118389-118427.

... [3] Achituve, Idan, et al. "Inverse problem sampling in latent space using sequential Monte Carlo." arXiv preprint arXiv:2502.05908 (2025).

... [4] Janati, Yazid, et al. "A Mixture-Based Framework for Guiding Diffusion Models." Forty-second International Conference on Machine Learning. 2025.

... [5] Qinsheng Zhang and Yongxin Chen. Fast sampling of diffusion models with exponential integrator. arXiv preprint arXiv:2204.13902, 2022.

**Questions:**

- Figure 1: can the authors clarify how the sample are being generated? Namely, is it one run of the algorithm and what is being plotted are the ensemble of particles?
- Lines 234–236: The claim that each particle in Blade has its own associated target distribution is unclear. Why would this not also apply to other ensemble-based samplers such as EKS or ALDI?
- Figure 3: How is the maximum rank defined in this context? What explains the apparent linear relationship between the rank and the step size? The figure caption mentions "accumulated rank" but its purpose and interpretation are unclear; what insight does plotting the accumulated rank provide about the algorithm’s behavior?

---

> ### Author Response · Authors · 2025-12-01
> **Response to "motivation of the method"**
>
> We thank the reviewer for the feedback. We agree that adding a bit more context about preconditioner choice improves clarity. The use of the covariance-preconditioned Langevin dynamics has well-established theoretical and practical justification beyond matrix inversion.
> 1. Because of statistical linearization, the potential function for our likelihood is quadratic. In this case, it is known that covariance preconditioning makes the dynamics affine-invariant and is asymptotically optimal. (See Sec. 3.2 in [6] and Lemma 4 in [7])
> 2. The intuition of covariance as preconditioner: we take larger movement in directions with large variance (wider distribution) and smaller movement in directions with small variance (narrow distribution). If standard Langevin dynamics is analogous to gradient descent, preconditioned Langevin dynamics is analogous to Newton's method or Adam optimizer with adaptive step sizes.
>
>
> To clarify this in the paper, we add the following to Sec. 3.1 right before Eq. (6):
> "
> Our starting point is the covariance-preconditioned Langevin dynamics with the large particle limit, which is known to have improved conditioning and convergence under quadratic potentials [6,7].
> "
>
> [6]: Reich, Sebastian, and Simon Weissmann. "Fokker--Planck particle systems for Bayesian inference: Computational approaches." SIAM/ASA Journal on Uncertainty Quantification 9.2 (2021): 446-482.
>
> [7]: Garbuno-Inigo, Alfredo, Nikolas Nüsken, and Sebastian Reich. "Affine invariant interacting Langevin dynamics for Bayesian inference." SIAM Journal on Applied Dynamical Systems 19.3 (2020): 1633-1658.

---

> ### Author Response · Authors · 2025-12-01
> **Response to "evaluation of the method"**
>
> **Summary**:
> - The reference cited by the reviewer uses only 4 numerical integration steps and explicitly states that "82 integration steps...would be expensive to differentiate through repeatedly." Our setup follows the more challenging InverseBench specification [9] requiring hundreds of steps, which is a well-established benchmark for derivative-free methods [8,9].
> - Navier-Stokes provides the right evaluation framework—probabilistic metrics (CRPS, SSR) from weather/ocean data assimilation—to assess our method's core contribution of calibrated posterior sampling as shown in Table 1.
>
>
> **Supporting details**:
> The reference "Score-based data assimilation" cited by the reviewer considers only 4 numerical integration steps, which substantially simplifies the gradient challenge. The reference explicitly mentioned that "82 integration steps of the forward Euler method, which would be expensive to differentiate through repeatedly, as required by gradient-based data assimilation approaches." In contrast, we follow the more challenging setup in [9], which requires hundreds of numerical steps. Typical real-world fluid solvers would take even more steps. Naively backpropagating through such long PDE rollouts becomes unreliable due to the sensitivity of the system [12,13].
>
> Moreover, the Navier-Stokes problem is a well-established benchmark problem for derivative-free methods [8,9], directly connecting to data assimilation problem in weather and oceanography [10,11], where probabilistic verification is standard. In this setting there are well-established scores like CRPS and the spread–skill ratio (SSR), that evaluate posterior samples probabilistically, not just pointwise error. This lets us assess the central aim of our method (calibrated posterior sampling) in a domain where the scoring rules are meaningful and widely used.
>
>
> **On black hole imaging and image restoration**: We include these experiments only in the appendix as complementary evidence of generality. These benchmarks emphasize deterministic reconstruction metrics and do not provide any probabilistic evaluations so they are not the primary setting for demonstrating the value of Blade.
>
> **Runtime**:
> - Blade produces a calibrated ensemble, not a single reconstruction. Producing 512 posterior samples inevitably requires more computation than a single gradient-based point estimate. The appropriate comparison is with methods offering similar posterior coverage. Fig. 14 includes all the appropriate baselines for our setting. and we see that all the runtime costs are comparable.
> - Additionally, the forward model itself (a PDE solver requiring hundreds of numerical steps) is about 900x more expensive (See Fig.6 in [9]) than common image corruption operators, which increases the runtime regardless of the algorithm used.
>
>
> [8]: Iglesias, Marco A., Kody JH Law, and Andrew M. Stuart. "Ensemble Kalman methods for inverse problems." Inverse Problems 29.4 (2013): 045001.
>
> [9]: Zheng, Hongkai, et al. "InverseBench: Benchmarking Plug-and-Play Diffusion Priors for Inverse Problems in Physical Sciences." The Thirteenth International Conference on Learning Representations.
>
> [10]: Park, Seon Ki, and Liang Xu, eds. Data assimilation for atmospheric, oceanic and hydrologic applications (Vol. II). Berlin, Heidelberg: Springer Berlin Heidelberg, 2013.
>
> [11]: Morcrette, Jean-Jacques, CH JAKOB, and J. TEIXEIRA. "European Centre for Medium range Weather Forecasts." Numerical Modeling of the Global Atmosphere in the Climate System 550 (2000): 263.
>
> [12]: Lea, Daniel J., Myles R. Allen, and Thomas WN Haine. "Sensitivity analysis of the climate of a chaotic system." Tellus A: Dynamic Meteorology and Oceanography 52.5 (2000): 523-532.
>
> [13]: Chandramoorthy, Nisha, et al. "Toward computing sensitivities of average quantities in turbulent flows." arXiv preprint arXiv:1902.11112 (2019).

---

> ### Author Response · Authors · 2025-12-01
> **Response to "Correctness"**
>
> We thank the reviewer for pointing out this ambiguity in our notation. We clarify that the noise in Eq. (11) is in $\mathbb{R}^J$ (not $\mathbb{R}^n$). So in general for $\sqrt{C_t}\in \mathbb{R}^{n\times J}$, we have $\sqrt{C_t}dw_t \in \mathbb{R}^n$. For single-particle case $J=1$, $\sqrt{C_t}dw_t$ is in $\mathbb{R}^n$ as intended.
>
> This is a known efficient implementation of the square root of the covariance matrix that can be found in [14]. We have added the clarification after Eq. (11) to prevent this confusion in the revised manuscript.
>
> [14]: Garbuno-Inigo, Alfredo, et al. "Interacting Langevin diffusions: Gradient structure and ensemble Kalman sampler." SIAM Journal on Applied Dynamical Systems 19.1 (2020): 412-441.

---

> ### Author Response · Authors · 2025-12-01
> **Response to "Missing references and related work"**
>
> We thank the reviewer for these valuable references. We have added both references to the "Diffusion-based Split Gibbs Sampling" paragraph in the related work section.
>
> Regarding Lines 90–91, we clarify that this statement is made specifically in the context of derivative-free algorithms, as indicated by the leading sentence of that paragraph. Reference [3], while addressing nonlinear inverse problems, explicitly relies on likelihood gradients (see Eq. (7) in [3]). To improve the clarity and precision, we have revised this statement in the line 88-90 in the updated manuscript

---

> ### Author Response · Authors · 2025-12-01
> **Response to typos and minor issues**
>
> **Typos**
> - We have corrected the typos in line 209.
> - Time horizon $t^\dagger, t^*$ refers to the time duration over which we evolve the respective dynamical systems defined in Eq. (11) and Eq. (2).
> - Regarding Line 86, we respectfully maintain the original phrasing "sampling for posterior inference" as we believe it more precisely describes inference in a probabilistic inversion context, which is central to our problem formulation.
>
> **$\beta(t)$ in Eq.(2)**
> We thank the reviewer for this comment. The coefficient $\beta(t)$ appears in a generalized form of the SDE that preserves the marginal distribution while allowing additional flexibility in the diffusion term.
>
>
> Our formulation is a direct generalization of the appendix D in [5] with the same theoretical argument. Specifically, Fokker Plank equation for Eq. (2) reads
>
> $$
> \begin{align}
> &\frac{\partial}{\partial t} p_t(x_t) \\
> &= - \nabla \cdot \left(p_t(x_t)\left(- (2\dot{\sigma}(t)\sigma(t)+\beta(t))\nabla_{x_t}\log p_t(x_t) \right) \right) - (\dot{\sigma}(t)\sigma(t)+\beta(t))\nabla_{x_t} \cdot \nabla_{x_t} p_t(x_t) \\
> & = - \nabla \cdot \left(p_t(x_t)\left(- \dot{\sigma}(t)\sigma(t)\right)\nabla_{x_t}\log p_t(x_t) \right)
> \end{align},
> $$
> where $\beta(t)$ terms cancel in the second equality, showing that the choice of $\beta(t)$ does not affect the marginal distribution.
>
> We've also corrected a typo in Eq. (2), which now reads
> $$
> dx_t=-(2\dot{\sigma}(t)\sigma(t)+\beta(t))\nabla_{x_t}\log p(x_t;\sigma(t))dt + \sqrt{2\dot{\sigma}(t)\sigma(t) + 2\beta(t)}d\bar{w}_t
> $$
>
>
> [5] Qinsheng Zhang and Yongxin Chen. Fast sampling of diffusion models with exponential integrator.

---

> ### Author Response · Authors · 2025-12-01
> **Response to questions**
>
> > Figure 1: ... how the sample are being generated?....
>
> In Figure 1, for ensemble methods including Blade, EKS, EnKG, the samples are generated in one run because they naturally produce the ensemble prediction. For the other methods like SCG, DPG, we simply perform repeated runs independently to collect enough samples.
>
> > Lines 234–236: The claim that each particle in Blade
>
> In Blade, the likelihood step for each particle evolves according to a particle-specific target potential. As shown in Eq. (6), the potential driving the update for particle $z^{(j)}$ depends on its own splitting variable $x^{(j)}$, which is sampled from denoising diffusion process and is therefore 1) different across particles and 2) changes over iterations. In contrast, EKS or ALDI is essentially a standalone preconditioned Langevin dynamics whose target potential is fixed.
>
> In contrast, methods such as EKS or ALDI simulate a single preconditioned Langevin dynamics whose target potential is fixed throughout the iterations and shared across all particles. Their particles interact through the ensemble covariance but the underlying target potential is the same for all particles and does not depend on any particle-specific auxiliary variable.
>
> > Figure 3: How is the maximum rank defined in this context? ...
>
> **Purpose of Figure 3**
> Many ensemble methods like EKI and EKS are known to remain confined to a low-dimensional subspace, which sometimes causes problems like particle collapsing or insufficient posterior coverage. Figure 3 is intended to assess whether Blade suffers from the same limitation by tracking the dimensionality of explored space over iterations.
>
> **Definition of max possible rank**
> At iteration $k$, we form the matrix whose columns are the concatenated particles from all iterations up to $k$: $[X^{0}, Z^{0}, \dots, X^{(k)}, Z^{(k)}]$. The accumulated rank is the rank of this matrix. The maximum possible rank is the total number of column vectors up to iteration $k$, which is `number of iterations x ensemble size`. This corresponds to the hypothetical case in which all ensemble states across all iterations are linearly independent.
>
> **Insights from Figure 3**
> The near-linear increase in accumulated rank indicates that the ensemble keeps exploring new directions over iterations. Thus the space spanned by the ensemble does not stagnate. The fact that accumulated rank reaches the ambient dimension within 35 steps shows that Blade has sufficient posterior exploration without the low-dimensional confinement that limits other ensemble methods.
>
> We have revised Remark 1 to better explain the experiment in Figure 3.

---

### Author Response · Authors · 2025-12-01
**General response**

We thank all reviewers for the detailed reviews and constructive feedback that have helped strengthen our paper. We are encouraged that reviewers found our work "well-written" (wxbH), "theoretically grounded" (jPCJ) with "extensive empirical evaluations" and "convincing evidence" (wxbH, jPCJ, vs1A).

All changes in the revised manuscript are marked in blue. We summarize the main updates in the updated manuscript as follows:
- **Additional baselines**: Figure 1 and Table 1 now include the suggested baselines: Feynman-Kac diffusion steering (wxbH), EKS w/ DP (jPCJ), localized EKS w/ DP (jPCJ), CDM-Cat (wxbH). The implementation details are added to Appendix C.2.3.
- **Enhanced theoretical exposition** (jPCJ): Added Remark 2 and expanded Remark 3 to better clarify our theoretical results. Moved expanded discussion of assumptions from appendix to main text (Lines 323-331).
- **Improved clarity**: Revised Remark 1 to better explain insights from Figure 3 (NR9p). Added context motivating covariance-preconditioned Langevin dynamics (Lines 205-207).

As a final remark, Blade is a novel diffusion-based derivative-free inference algorithm that produces calibrated samples with theoreical guarantees that explicitly control errors from forward model and diffusion model. We highlight two key aspects of the paper.
- Blade addresses the challenging derivative-free inversion problem in a principled manner. It produces **well-calibrated** posterior samples where existing methods fail. This has important implications for scientific domains where uncertainty quantification is critical.
- Our theoretical analysis strictly generalizes the theory in prior work (e.g. PnPDM) to a more realistic and technically challenging regime.
    - We consider an interacting particle system with state-dependent preconditioner within the split Gibbs framework, which is more complex than the standard Langevin dynamics in the piror work. This requires non-trivial technical extensions.
    - Our analysis setup is more realistic and practically relevant by considering finite time execution of the dynamics as well as the forward model error. Theorem 2 explicitly shows how the algorithm remains stable under forward model error and how the finite time horizons affect convergence, which are not addressed in the prior work.

---

### Meta-Review · Area_Chair_7F6a · 2026-01-06

**Summary:**

The submission received mixed reviews. Two reviewers were overall positive, but their assessments are quite sparse and do not provide enough technical detail to strongly support acceptance. Reviewer NR9p was very negative (rating 2), focusing on motivation and evaluation; these points appear to be addressed reasonably well in the rebuttal. The remaining key issues, however, come from reviewer jPCJ and are more fundamental.

In particular, concerns remain about (1) the validity of the error bound in Theorem 2, (2) the limited novelty of the convergence proof relative to existing frameworks, (3) the practicality of the linearization assumption, and (4) missing comparisons against EKS and ALDI with localization. The authors partially addressed (4) by adding comparisons (e.g., EKS+DM), which is helpful, but several theoretical aspects remain insufficiently resolved. The response does not clearly articulate what is substantively new in the convergence analysis, and the empirical boundedness evidence used to justify Assumption 2 does not provide a theoretical guarantee. Moreover, much of the theoretical development applies to the split Gibbs sampler with $\rho>0$, which departs from the original posterior sampling objective; the paper does not adequately analyze or bound the gap between the original posterior and the distribution induced by the proposed sampling scheme. Given these unresolved core theoretical questions and some remaining evaluation gaps, I lean toward a weak reject.

**Reviewer Concerns:**

.

**Reviewer Scores:**

.

---

### Decision · Program_Chairs · 2026-01-26

Reject